# End-to-End Compression for Tabular Foundation Models

**Guri Zabërgja** [* 1]  **Rafiq Kamel** [* 2]  **Arlind Kadra** [2 3]  **Christian M. M. Frey** [2]  **Josif Grabocka** [2]

## Abstract

The long-standing dominance of gradient-boosted decision trees for tabular data has recently been challenged by in-context learning tabular foundation models. In-context learning methods fit and predict in one forward pass without parameter updates by leveraging the training data as context for predicting on query test points. While recent tabular foundation models achieve state-of-the-art performance, their transformer architecture based on the attention mechanism has quadratic complexity regarding dataset size, which in turn increases the overhead on training and inference time, and limits the capacity of the models to handle large-scale datasets. In this work, we propose TACO, an end-to-end tabular compression model that compresses the training dataset in a latent space. We test our method on the TabArena benchmark, where our proposed method is up to 94x faster in inference time, while consuming up to 97% less memory compared to the state-of-the-art tabular transformer architecture, all while retaining performance without significant degradation. Lastly, our method not only scales better with increased dataset sizes, but it also achieves better performance compared to other baselines.

## 1. Introduction

Tabular data has ever since been a prevailing data structure in a plethora of domains due to its flexibility towards storing heterogeneous feature types in relational databases (Borisov et al., 2021; Peleska & Sír, 2024; Jiang et al., 2026). In the last decade, the overwhelming success of foundation models catalyzed a parallel agenda for tabular learning (Van Breugel & Van Der Schaar, 2024) next to traditional

---
[*]Equal contribution  [1]Department of Computer Science University of Freiburg  [2]Department of Computer Science, Technical University of Nuremberg  [3]DarkNeurons. Correspondence to: Guri Zabërgja <zabergjg@informatik.uni-freiburg.de>, Rafiq Kamel <rafiq.kamel@utn.de>.

*Proceedings of the 43rd International Conference on Machine Learning*, Seoul, South Korea. PMLR 306, 2026. Copyright 2026 by the author(s).

learning approaches like gradient-boosted decision trees (Chen & Guestrin, 2016; Ke et al., 2017; Prokhorenkova et al., 2018), random forests (Breiman, 2001), or feed-forward models (Kadra et al., 2021; Gorishniy et al., 2021; Kadra et al., 2024; Holzmüller et al., 2025; Gorishniy et al., 2025). The main essence of tabular foundation models relies in learning across a broad distribution of synthetic and real datasets, which enables a fast adaptation to an unseen dataset at inference time without task-specific retraining (Hollmann et al., 2023; Qu et al., 2025; Hollmann et al., 2025). A compelling paradigm for prediction on tabular data is in-context learning (Hollmann et al., 2023), where at inference time, the model conditions on a dataset's training split and produces predictions for test points via a single forwards pass, which effectively amortizes the learning procedure itself into the model's weights.

Recent tabular foundation models have demonstrated that in-context predictors are on par or even surpass strong classical baselines on tabular benchmark datasets. In particular, recent benchmark efforts (Zabërgja et al., 2025; Erickson et al., 2025) have positioned tabular foundation models as state-of-the-art for tabular classification. Prominent representatives of this paradigm are, e.g., TabPFN (Hollmann et al., 2025), TabICL (Qu et al., 2025), and TabDPT (Ma et al., 2024). These models share a common architectural pattern based on decoder-only transformers that attend over a tokenized representation of the training split and predict labels for test points conditioned on that context in a single forward pass. However, the architectural choice also induces a fundamental scaling bottleneck. The inference costs scale quadratically with the size of the conditional context, resulting in $\mathcal{O}(N^2)$ inference-time bottleneck in the number of training instances $N$. If Key-Value (KV) caching is used, the inference-time bottleneck can be reduced to $\mathcal{O}(N)$. More concretely, when the training split contains $N$ rows with $M$ features, many implementations effectively induce memory and compute that scale with the number of cells within the tabular data, i.e., $N \times M$, which quickly reaches hardware budget restrictions in practical scenarios (Zhong et al., 2024). Common workarounds are to train only on feasible context sizes that force models to include a subsampling of rows/features (Ye et al., 2026; Eisenschlos et al., 2021) or to restrict the training only to small and mid-size tables (Hollmann et al., 2023; 2025).

Consequently, the predictive performance degenerates on large, information-enriched datasets (Qu et al., 2025). Another drawback stems from the inference latency and cost itself, where tabular foundation models are only attractive and impactful in practical scenarios if the forward pass is tractable at realistic $N$ rows and $M$ features.

To address this central bottleneck, we propose TACO, a novel modular architecture for fast context compression that can be integrated with any decoder-only tabular foundation model to linearly reduce its inference complexity regarding $N$. Our approach introduces a learned compressor that reduces the training context before prediction, thereby decoupling predictive quality from the raw size of the conditional context table. Under the hood, TACO maps an input dataset into a compact representation, and performs in-context prediction using only the compressed context. Intuitively, our compressor module learns a condensed embedding capturing $K \ll N$ prototypical patterns of the tabular data, and the predictor module produces test-point predictions conditioned on the compressed representation. Therefore, TACO directly tackles the attention scaling problem as the compressor reduces the effective context size by a factor $N/K$, denoting the compression rate, resulting in a proportionally reduction of memory and compute costs. TACO reframes tabular in-context inference by learning *how to compress*, i.e., by learning data representations that remain predictive under strict inference-time budgets. Notably, our end-to-end pipeline enables a principled accuracy-latency trade-off via the compression rate. This allows tuning resource use under deployment constraints and allows in-context tabular predictors to scale to datasets from real-world domains.

In a thorough evaluation, we demonstrate that it is possible to compress the context down to $1\%$ while incurring no statistically significant deterioration of predictive performance relative to the uncompressed foundation model baseline.

Overall, our contributions can be summarized as follows:

- We introduce TACO, an end-to-end context compression method for tabular in-context learning.

- We demonstrate empirically that TACO offers a $94\times$ speedup during inference and $98\%$ memory savings with no significant performance degradation at only $1\%$ of the original context size.

- Lastly, we derive a simple chunking-and-stitching strategy that enables ingestion of datasets with over one million rows.

## 2. Related Work

**Deep Learning for Tabular Data.** Tabular data has traditionally been dominated by tree-based methods (Chen & Guestrin, 2016; Ke et al., 2017; Prokhorenkova et al., 2018), while numerous novel deep learning architectures (Popov et al., 2020; Arik & Pfister, 2021; Gorishniy et al., 2021; Ye et al., 2025) have been proposed, they often fall short in achieving better predictive performance (Grinsztajn et al., 2022; Shwartz-Ziv & Armon, 2022; McElfresh et al., 2023). However, recently simple feed-forward networks (Kadra et al., 2021; Holzmüller et al., 2025) and efficient ensembles of feed-forward neural networks (Gorishniy et al., 2025) have managed to outperform traditional methods such as gradient-boosted decision trees, as verified by multiple studies (Erickson et al., 2025; Zabërgja et al., 2025).

**Tabular Foundation Models (TFMs).** The recent success of foundation models in the natural language (Radford et al., 2019; Brown et al., 2020), and computer vision domain (Siméoni et al., 2025), has shifted the attention of practitioners in developing tabular foundation models to replicate the success in the tabular data domain (Van Breugel & Van Der Schaar, 2024). TabPFN (Hollmann et al., 2023) is among the initial works that use a transformer-based model trained on synthetic data that achieves state-of-the-art prediction results in small to medium-sized datasets. Later versions of the work enhance the initial architecture by alternating column and row-wise attentions (Hollmann et al., 2025). Alternative approaches (Qu et al., 2025), use a more efficient architecture to handle datasets of larger sizes. Recently, TabDPT (Ma et al., 2025) and ConTextTab (Spinaci et al., 2025) have shown that it is possible to build a tabular foundation model relying solely on real data and achieve competitive performance. Lastly, training on a mixture of synthetic and real data achieves state-of-the-art predictive performance (Grinsztajn et al., 2025).

**Compression Methods.** While transformer-based architectures achieve state-of-the-art results compared to traditional methods, they suffer from a squared complexity with regards to the number of input instances, which results in a higher latency overhead for training and inference. To that end, there exists prior work that aim to improve the efficiency of the current tabular foundation models. MotherNet (Müller et al., 2023) trains a meta-learned transformer hypernetwork that predicts the weights of a simple feed-forward neural network to reduce the inference time, effectively distilling the parameters of the large transformer architecture in a simple per-dataset neural network. Additionally, TabFlex (Zeng et al., 2025) incorporates linear attention to reduce training and inference overhead, and effectively scale to larger datasets. To the best of our knowledge, there is no published methodology for compressing the training context of tabular foundation models in an end-to-end manner.

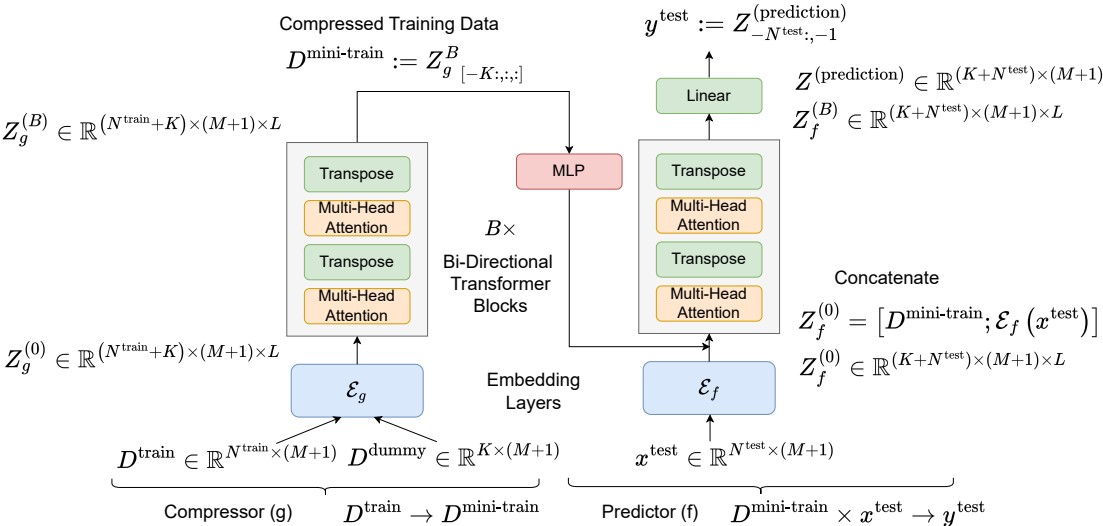

*Figure 1.* The architecture for training a joint end-to-end compressor and a predictor, with each model having its own parameters.

## 3. TACO: Tabular Compression

### 3.1. Tabular Foundation Models

Our problem is tabular prediction, which consists of predicting the target variable $y \in \mathbb{R}^N$ of a feature vector $x \in \mathbb{R}^{N \times M}$, where $N$ is the number of training data points, and $M$ is the number of features. We define a tabular dataset $D := \{(x_i, y_i)\}_{i=1}^N$ as a matrix $D \in \mathbb{R}^{N \times (M+1)}$, obtained by concatenating the features and target variables into $(M + 1)$-dimensional vectors. A prior over datasets, also referred to as the meta-training collection of datasets, is defined by $(D^{\text{train}}, D^{\text{test}}) \sim p(D)$. A batch of test data points is consequently given by $(x^{\text{test}}, y^{\text{test}}) \in D^{\text{test}}$. A foundation model with parameters $\theta$ is designed as an in-context prediction model $f(x^{\text{test}}, D^{\text{train}}; \theta) : \mathbb{R}^{N^{\text{test}} \times M} \times \mathbb{R}^{N^{\text{train}} \times (M+1)} \to \mathbb{R}^{N^{\text{test}}}$ where, the output corresponds to class scores for the test points. We train the foundation model's parameters through the following log-likelihood minimization over the prior of datasets $p(D)$, by minimizing a loss function $\mathcal{L} : \mathbb{R} \times \mathbb{R} \to \mathbb{R}_+$, such as cross-entropy or least-squares error.

$$\arg\min_{\theta} \; \mathbb{E}_{\substack{(D^{\text{train}}, D^{\text{test}}) \sim p(D) \\ (x^{\text{test}}, y^{\text{test}}) \sim p_{D^{\text{test}}}}} \mathcal{L}\left(y^{\text{test}}, f(x^{\text{test}}, D^{\text{train}}; \theta)\right) \quad (1)$$

The architecture of state-of-the-art tabular foundation models is a bi-directional transformer, first proposed by (Lorch et al., 2022) and later popularized by recent models (Hollmann et al., 2023; Qu et al., 2025; Hollmann et al., 2025). First of all, each cell of a table $D \in \mathbb{R}^{N \times (M+1)}$ with $N$ rows, $M$ features, and a target variable, is projected to an $L$-dimensional space with an encoder layer $\mathcal{E}_f(D)$ :

$\mathbb{R}^{N \times (M+1)} \to \mathbb{R}^{N \times (M+1) \times L}$, yielding a latent representation $Z_f^{(0)}$. Then, a sequence of two multi-head attention blocks (Vaswani et al., 2017) is applied, in the form of a row-wise attention across $N$ cells for each column, and a column-wise attention across $M + 1$ cells for each row (Hollmann et al., 2025). The latent representation of a dataset after each transformer block is denoted by $Z_f^{(b)} \in \mathbb{R}^{N \times (M+1) \times L}, b \in \{1, \ldots, B\}$.

### 3.2. End-to-End Compressor and Predictor Networks

The bi-directional transformers architectures suffer from an $\mathcal{O}\left(\left(N^{\text{train}}\right)^2 \times M + M^2 \times N^{\text{train}}\right) = \mathcal{O}\left(\left(N^{\text{train}}\right)^2 \times M\right)$ inference complexity in the default variant, given that $M \ll N^{\text{train}}$ in most practical scenarios. If KV caching is used, the inference complexity is reduced to $\mathcal{O}\left(N^{\text{train}} \times M\right)$.

In this paper, we speed up the inference of tabular foundation models by compressing the training set $D^{\text{train}}$ into a $D^{\text{mini-train}}$, where the latter has $K \ll N^{\text{train}}$ rows. Therefore, the inference cost for the attention layers is reduced by a factor of $\left(\frac{N^{\text{train}}}{K}\right)^2$ in the default transformer variant, and by $\left(\frac{N^{\text{train}}}{K}\right)$ in the case of KV caching.

Our methodology consists of two end-to-end transformer architectures, a compressor $g : D^{\text{train}} \to D^{\text{mini-train}}$ with parameters $\phi$ that computes a lower rank representation of the training data, and a predictor $f : D^{\text{mini-train}} \times x^{\text{test}} \to y^{\text{test}}$ with parameters $\theta$ that takes as an input the compressed training set and the feature matrix of a test batch, to predict the target variable $y^{\text{test}}$ of the test features.

We feed the compressor network $g$ with the original training data $D^{\text{train}} \in \mathbb{R}^{N^{\text{train}} \times (M+1)}$, and a dummy table $D^{\text{dummy}} \in \mathbb{R}^{K \times (M+1)}$, where $K \ll N^{\text{train}}$. Such dummy rows are initialized randomly from a subset of $D^{\text{train}}$. Throughout the transformer block, the network is trained to compress $D^{\text{train}}$ into the latent representation of the dummy rows, yielding a final representation $D^{\text{mini-train}} \in K \times (M+1) \times L$. To differentiate between the real and dummy cells in the attention and feed-forward layers of the compressor network, we mask the $(M+1)$-th column of the input dummy matrix, corresponding to the target variable, with a special placeholder value.

The output of the compressor is fed into a predictor network $f$, which is a standard tabular transformer model (Hollmann et al., 2025). We concatenate the output of the compressor $D^{\text{mini-train}}$ with the embedding of a test batch $\mathcal{E}_f(x^{\text{test}})$ as the initial tensor to the predictor's attention blocks $Z_f^{(0)}$. The predictor's transformer network fuses the latent representation of the compressed training set and the embedded test batch to predict the target variable of the test batch. A visual representation of our novel tabular compression architecture is shown in Figure 1.

The compressor $g$ and the predictor $f$ are meta-learned jointly in an end-to-end manner as follows:

$$\underset{\theta,\phi}{\arg\min} \ \mathbb{E}_{\substack{(D^{\text{train}},D^{\text{test}})\sim p(D) \\ (x^{\text{test}},y^{\text{test}})\sim p_{D^{\text{test}}}}} \mathcal{L}\left(y^{\text{test}}, f\left(x^{\text{test}}, g\left(D^{\text{train}};\phi\right);\theta\right)\right)$$
(2)

As a result of the end-to-end learning, the compressor is incentivized to produce a low-rank training set representation $D^{\text{mini-train}}$ that maximally improves the performance of the predictor. At the same time, the predictor is trained to fuse the $L$-dimensional cells of the compressed training data and the test batch embedding into a joint representational space.

## 4. Experimental protocol

This section describes the experimental protocol used to train and evaluate our models. We first detail the training setup, including model architecture, optimization strategy, and the prior used during pretraining. We then introduce the evaluation setup, covering the datasets, metrics, caching mechanisms, and terminology that will be used throughout the empirical analysis. Based on this setup, we subsequently formulate a set of research insights that structure the results presented in the following section.

### 4.1. Training Setup

We begin by describing the training configuration shared across all experiments. Unless stated otherwise, all models are trained using the same protocol to ensure stable optimization and fair comparison.

**Model Architecture and Hyperparameters.** For both the compressor and predictor modules in TACO, we use the TabPFNv2 (Hollmann et al., 2025) architecture with 2D attention over rows and columns. Each module comprises 12 attention layers, 6 attention heads, and an embedding dimension of 192, for roughly 7M parameters per module. We additionally insert a two-layer residual MLP between the compressor and predictor.

**Synthetic Prior.** Following prior work (Hollmann et al., 2023; Qu et al., 2025), we train our models initially on synthetically generated datasets produced by structural causal models (SCMs), including variants where the causal mechanisms are tree-based. This synthetic prior is designed to expose the model to a broad range of feature types, dependency structures, and label-generating mechanisms that are representative of real-world tabular data. We adopt the synthetic prior released with TabICLv1 (Qu et al., 2025) and use its open-source implementation throughout.

**Optimization and Training Length.** Following prior work (Qu et al., 2025), during training, we fix the sequence length to 1024 and sample the number of features uniformly at random between 2 and 100. We train for 80K optimization steps with a global batch size of 1024, yielding approximately 82M synthetic datasets. To further study the robustness and adaptability of TACO beyond the original 80k-step synthetic pretraining regime, we explored additional continuation-training stages inspired by the curriculum-style setup used in TabICL. In particular, we progressively increased the supported context lengths across multiple synthetic continuation stages, first extending training to 81k optimization steps using sequence lengths between 1k and 40k rows, and subsequently to 81.025k steps using substantially longer sequences between 40k and 60k rows. Throughout these stages, we retained the same model architecture, optimization setup, and compression configuration.

Starting from the final synthetic continuation checkpoint, we then performed an additional real-data adaptation stage, extending to 91k total optimization steps, using real tabular datasets. Following the training procedure proposed in TabDPT (Ma et al., 2024), we constructed self-supervised prediction tasks by bootstrapping rows and features from real datasets and randomly selecting one feature as the prediction target. For continuous targets, values were discretized into at most 10 bins to remain compatible with the classification-style training setup used throughout synthetic pretraining. During this phase, we reduced the maximum context length to 20k rows while keeping the remaining training setup unchanged. To alleviate GPU memory constraints, we use gradient accumulation with a microbatch size of 16. We optimize using AdamW and train with mixed-precision arithmetic. We warm up the learning rate to $1 \times 10^{-4}$ and subsequently apply a cosine anneal-

ing schedule (Loshchilov & Hutter, 2017). We additionally employ gradient clipping with a maximum norm of 1.0 and use weight decay of $1 \times 10^{-2}$. For each of our models, we employ distributed training across 8 NVIDIA H100 GPUs with 95 GB of VRAM each. Pretraining took approximately 20 days to complete. For reproducibility, we open-source our code at: https://github.com/machinelearningnuremberg/TACO.

## 4.2. Evaluation Setup

In this subsection, we describe the evaluation protocol used to assess the scalability and predictive performance of TFMs. This setup defines the datasets, metrics, and conventions referenced throughout the results section.

**Evaluation Datasets and Metrics.** We evaluate all models and baselines on the classification subset of TabArena (Erickson et al., 2025). For most experiments, we restrict this evaluation to the same set of 26 datasets used in the TabPFN v2.0 evaluation (Hollmann et al., 2023). Among these datasets, 20 are binary classification tasks, and 6 are multiclass. For the experiments corresponding to Insight 6, we use all 36 classification datasets with fewer than 10 classes in TabArena, of which 8 are multiclass.

We follow the same data preprocessing pipeline as TabPFN v2.0 (Hollmann et al., 2023) and use the predefined cross-validation folds from TabArena (Erickson et al., 2025). Model performance is evaluated using ROC-AUC for binary classification tasks and ROC-AUC (one-vs-one) for multiclass tasks (Junge & Dettori, 2018). For simplicity, we refer to both metrics as ROC-AUC throughout this section.

**Terminology.** Throughout this section, we use the term *compression rate* $r$ to denote the ratio between the number of rows $K$ in the compressed training representation produced by the compressor and the number of rows $N^{\text{train}}$ in the original training set, i.e., $r = K/N^{\text{train}}$. We refer to *POT* as the *predictor-only transformer*, which consists solely of the predictor architecture used in TACO and is trained under the same data, optimization, and hyperparameter setup, but without any compression component; the predictor therefore operates directly on the full, uncompressed training data. *TACO* denotes the full model that combines a learned compressor with a predictor transformer of the same architecture as POT, trained jointly in an end-to-end manner. We use *TACO* ($r\%$) to indicate a configuration in which the compressor produces a compressed training representation with a compression rate of $r\%$.

## 4.3. Research Insights and Experiments

**Insight 1: TACO achieves up to 94× speedup during inference, without significant degradation in performance.** Initially, we construct a grid of synthetic datasets with

$N \in [5000, 50000]$ rows and $M \in [100, 1000]$ features. For each grid point, we evaluate both TACO with a compression rates of $r = 1\%$ and the plain counterpart predictor-only transformer (POT). We evaluate both methods with and without KV caching, and measure fit and prediction time. Where, fit time represents the preprocessing time.

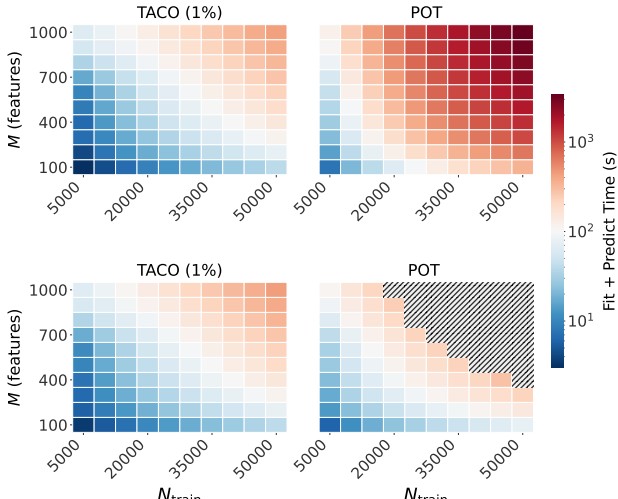

*Figure 2.* Fit + Predict time heatmap. **Top:** Predict times for TACO and POT, without using KV caching. **Bottom:** Predict times with KV caching. Black shaded regions indicate out-of-memory errors.

We consider three inference scenarios: (i) providing the full test set to the model in a single call, (ii) splitting the test set into 10 batches, and (iii) splitting the test set into 100 batches. For consistency, we fix the test set to 1,000 rows. Results for scenario (ii) are shown in Figure 2, while the remaining scenarios are reported in Appendix A.1.

In all experiments, KV caching applies exclusively to the predictor transformer module and is used identically for both TACO and the predictor-only transformer (POT). KV caching requires materializing and storing the key–value tensors of all attention layers, which introduces additional memory overhead compared to standard inference, where attention maps can be recomputed on the fly. In contrast, caching the output of the compressor in TACO incurs no additional memory cost, since the compressed context must be materialized regardless in order to be consumed by the predictor. Consequently, for TACO, KV caching refers solely to caching the predictor's attention states over the compressed context, while the compressor itself is executed only once and never recomputed. This distinction explains why KV caching is often infeasible for POT under tight memory budgets, yet remains practical when combined with TACO.

Figure 2 (Top) presents the heatmap without KV caching. Across all grid points, TACO is consistently faster, as indicated by the blue and light orange color throughout the

heatmap. Likewise, Figure 2 (Bottom) shows that TACO remains the faster method under KV caching, with performance advantages that are consistent across the entire grid. In contrast, POT with KV caching runs out of memory on 32 out of 100 datasets in the grid. These results also suggest that TACO can serve as an enabler of KV caching by making cached inference feasible under tighter memory budgets.

To further highlight the speedup of our approach, we simulate an "infinite test-batches" setting, which is common in industrial deployments (e.g., real-time fraud detection, ad click-through prediction, predictive maintenance) where predictions must be produced continuously for a stream of incoming data. We generate a synthetic dataset with 15,000 rows and 500 features, and evaluate inference over 100 sequential test batches of 50 rows each.

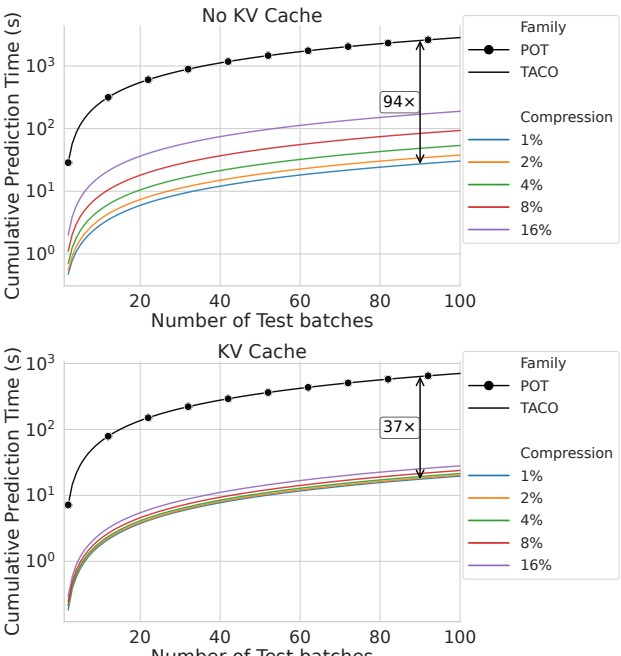

*Figure 3.* Cumulative prediction time for TACO at different compression rates $r$ compared to POT for 100 test batches. **Top:** Without using KV Caching, **Bottom:** Using KV Caching.

Figure 3 reports the cumulative prediction time (y-axis) as a function of the number of processed batches (x-axis). By compressing the context once and reusing it across batches, TACO delivers substantial gains after the first batch, which incurs a one-time cost to generate the compressed context (or to perform the initial fit in the KV caching setting), achieving speedups of up to $\sim 94\times$ at a 1% compression rate for the no KV caching setting, and up to $\sim 37\times$ for the KV caching setting. The gap in time between TACO and POT increases with dataset size.

Next, it is important to validate that the speedup gains and memory consumption reduction that TACO obtains do not

*Table 1.* Mean and standard deviation of TACO and POT over the TabArena datasets compared to recent state-of-the-art methods.

| Model | Mean ROC AUC ($\uparrow$) |
|---|---|
| TabICL | $0.866 \pm 0.103$ |
| TabPFN V2.0 | $0.866 \pm 0.103$ |
| POT | $0.862 \pm 0.101$ |
| TACO (r=1%) | $0.855 \pm 0.097$ |
| TACO (r=2%) | $0.857 \pm 0.098$ |
| TACO (r=4%) | $0.857 \pm 0.099$ |
| TACO (r=8%) | $0.858 \pm 0.100$ |
| TACO (r=16%) | $0.858 \pm 0.101$ |

*Note: Highlighted methods use open-weight checkpoints.*

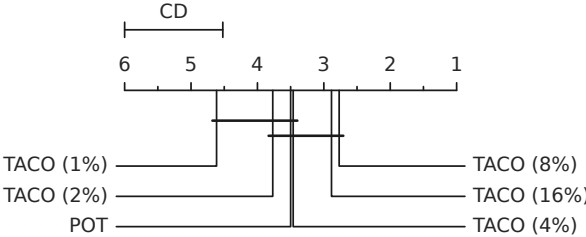

*Figure 4.* Critical difference (CD) diagram comparing predictor-only transformer (POT) and TACO. Average ranks are computed using Friedman–Nemenyi tests (Demšar, 2006) via `autorank` (Herbold, 2020b) ($\alpha = 0.05$). Bars connect methods without significant differences.

come at the cost of significant degraded predictive performance. To assess this trade-off, we compare TACO against POT on the TabArena benchmark (Erickson et al., 2025).

In Figure 4, we provide critical difference (CD) diagrams to investigate the method ranks and the statistical significance of the results. To build the CD diagrams, we use the autorank (Herbold, 2020a) package that runs a Friedman test with a Nemenyi post-hoc test, and a 0.05 significance level. The results indicate that **up to 1% compression rate, TACO provides a 94$\times$ speedup in inference time with no significant degradation in performance.**

Furthermore, in Table 1 we provide the mean performance and standard deviation of TACO with various compression rates, and POT against open-source checkpoints of state-of-the-art architectures, where, as observed, although POT and TACO have not been trained for as long as competitor methods, they achieve comparable performance.

Lastly, in Figure 5 we provide a comparison with the top-performing methods in the TabArena benchmark where fine-tuning, hyperparameter optimization, and ensembling are incorporated. As can be observed, TACO matches POT in performance and achieves a similar performance to TabICL. De-

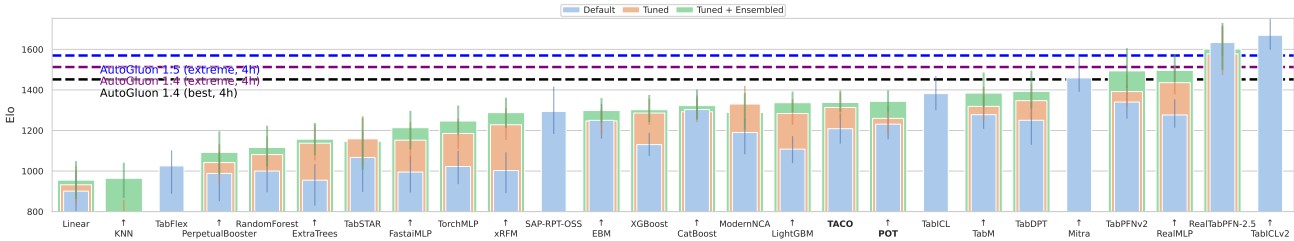

*Figure 5.* TabArena Elo comparison of TACO and POT against classical machine learning, deep learning, and tabular foundation model baselines under default, tuned, and tuned+ensembled evaluation settings. Our methods are highlighted in bold. Dashed horizontal lines denote AutoGluon reference systems reported by TabArena.

*Table 2.* Inference time and memory consumption for the predictor-only transformer (POT), versus TACO with different compression rates, without KV caching.

| Method | Fit Time | First Predict Time | Subsequent Predict Time | Fit Mem | Predict Mem | |
|---|---|---|---|---|---|---|
| | | | Mean ± Std | | First Predict | Subsequent |
| POT | 8.41s | 28.55s | 28.67s±0.05s | 36MB | 22.45GB | 22.45GB |
| TACO 1% | 8.22s | 29.68s | 306ms±18ms ($\times 93.6$) | 92MB | 22.69GB | 549MB ($-97.6\%$) |
| TACO 2% | 8.30s | 29.80s | 382ms±18ms ($\times 75.2$) | 92MB | 22.93GB | 845MB ($-96.3\%$) |
| TACO 4% | 9.86s | 30.71s | 544ms±17ms ($\times 52.7$) | 92MB | 23.42GB | 1.41GB ($-93.7\%$) |
| TACO 8% | 10.02s | 32.47s | 943ms±17ms ($\times 30.4$) | 92MB | 24.39GB | 2.56GB ($-88.6\%$) |
| TACO 16% | 9.20s | 35.36s | 1.91s±0.01s ($\times 15$) | 92MB | 26.34GB | 4.89GB ($-78.2\%$) |
| XGBoost | 3.14s | 154ms | 6.3ms±0.2ms | 8MB | 8MB | 8MB |
| AutoGluon Extreme | 30.2min | 1.32s | 947ms±10ms | 6.41GB | 8MB | 8MB |

Parentheses: $\times$ speed-up, $-\%$ memory reduction (vs. POT).

tailed reports of the performance distribution across datasets can be found in Section C.

**Insight 2: TACO consumes 98% less memory compared to the predictor-only transformer during prediction.** The compressor component of TACO inherits the same asymptotic complexity as the predictor-only transformer architecture; however, this is a one-time cost incurred during compression. While the predictor component of TACO also scales linearly as the predictor-only transformer, it scales with respect to the retained context size $K$ rather than the full dataset size $N$. Since $K \ll N$, its memory complexity is $\mathcal{O}(K \times M)$, yielding substantially lower memory usage in practice. We measure memory consumption for the same streaming-batch experiment shown in Figure 3 and report the results in Table 2 for the no KV cache setting.

The results show that using a $K$-sized compressed context yields substantial memory savings during prediction. Specifically, TACO achieves a 78.2% memory reduction for TACO (16%) and up to 97.6% for TACO (1%) in the no KV caching setting. While our fit-time memory footprint is $\sim 2.5\times$ higher than that of the predictor-only transformer in the no-KV-cache setting, **this is typically not a practical concern because fitting without KV caching is comparatively memory-friendly compared to the predict phase**. In contrast, fit-time memory becomes a bottleneck when KV

caching is enabled, since the model must store key and value tensors for every token, across all attention layers and heads. In this formulation, where each table cell corresponds to a token, this storage requirement can scale rapidly and become prohibitive. **In the KV caching setting, our model consumes 92% less memory compared to the predictor-only transformer**. For the KV cache results, we refer the reader to Table 4 in Appendix A.1.

**Insight 3: Joint training of the compressor and predictor is necessary.** To test whether the joint training of compressor and predictor is necessary, we train a variant of TACO in which the predictor is initialized with the TabPFN v2.0 (Hollmann et al., 2025) weights and kept frozen throughout meta-learning, while only the compressor parameters are optimized. We compare this variant to the standard TACO model under identical experimental conditions across a range of compression rates.

The results in Figure 6 verify that **jointly optimizing the predictor and compressor consistently leads to better performance compared to optimizing the compressor alone with a frozen predictor.** This highlights the importance of joint training: folding compressed representations into a predictor's representation space allows the predictor itself to adapt, rather than requiring the compressor to align with a fixed predictor.

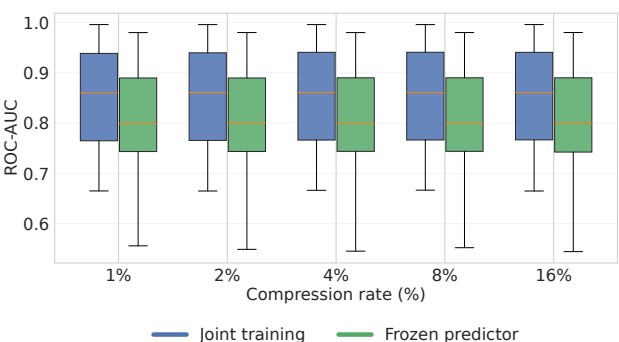

*Figure 6.* Distribution of test performances on the TabArena classification tasks, for the: **Left:** joint training of the compressor and predictor, **Right:** training only compressor, while keeping the predictor frozen.

**Insight 4: Meta-learning over various compression rates does not incur a performance loss compared to meta-learning at a fixed compression rate.** Multi-rate compression rate training enables a single TACO model to operate across multiple compression rates by sampling the compression rate uniformly from $\{1, 2, 4, 8, 16\}$ for each synthetic dataset during training. To assess whether this flexibility leads to performance degradation, we compare this *multi-rate model* to *rate-specific* TACO models, each trained using a single fixed compression rate $r$.

All models use the same architecture, training procedure, prior, and number of training datasets, and are evaluated at their corresponding compression rates.

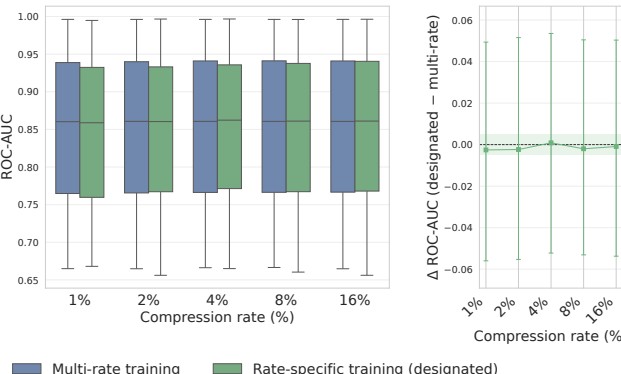

*Figure 7.* Comparison of dynamic compression rate training for the compressor, versus specific compression rate training. **Left:** Distribution of ROC-AUC test performances over all TabArena classification datasets, **Right:** ROC-AUC differences between specific and dynamic compression rate training; green-hued lines denote effects with **95% confidence intervals**.

As shown in Figure 7, performance differences between the multi-rate and rate-specific models are centered near zero across all compression rates, with no statistically significant advantage for rate-specific training at the 95% confidence level. This indicates that multi-rate compression rate train-

ing does not result in a measurable performance loss.

**Insight 5: TACO outperforms additional compression baselines.** We further compare **TACO** to two compression approaches on the TabArena benchmark: **kNN-based sampling**, where we form a training context subset by selecting $k$-nearest neighbors for each test point, increasing $k$ until the union reaches the target size and uniformly subsampling if needed to ensure the correct compression rate and roughly equal test-point contribution; and **random sampling**, where we subsample uniformly from the training set until we reach the compression rate.

We evaluate these compression baselines with the predictor-only transformer, comparing their performance across a range of compression rates. As shown in Figure 8, **TACO consistently and significantly outperforms both baseline methods across different compression rates ranging from 1% to 16%**. As the compression rate increases, the performance gap gradually narrows, since using more rows from the training set brings all methods closer to using the full training set. Moreover, TACO introduces no additional runtime and only a marginal memory overhead compared to the baselines, which is negligible relative to its memory savings over POT. Additional details on runtime and memory costs are provided in the Appendix A.2.

**Insight 6: TACO enables efficient chunking of large tabular datasets while preserving strong learned embeddings under extreme compression.** Current tabular foundation models often struggle to scale to large training sets because the memory cost of attention grows quickly with the number of rows. A key advantage of our end-to-end compression pipeline is that it does not require loading the full dataset into the compressor at once. Instead, we use an efficient *chunking-and-stitching* procedure that enables scalable compression and inference.

For example, given a dataset with $N = 10^6$ rows, we partition it into $N_C$ disjoint chunks of size $C = 10^4$, yielding $N_C = N/C = 100$ chunks. We then process chunks sequentially, by applying the compressor with a certain compression rate e.g. 1%, producing a compressed summary with $K_C$ rows (e.g., 100 rows per chunk). Finally, we *stitch* the compressed outputs by concatenating the per-chunk summaries into a single global context, which is provided as context to the predictor at test time.

Using this approach, we successfully perform prediction on the MetroPT-3 (Davari et al., 2021; Veloso et al., 2022) dataset, which contains approximately 1.5M rows and 15 features. For this dataset we use a simple 80%/20% time-based train/test split and a compression rate of 0.1%, compressing the training set from $\sim 1.2$M rows to only 1214 retained compressed rows. We compare against both the random and kNN baselines, and report results in Table 3.

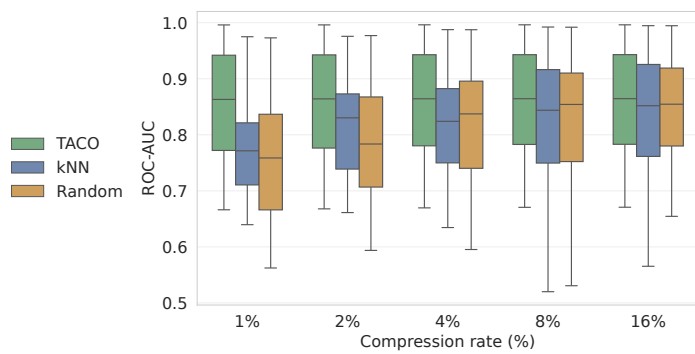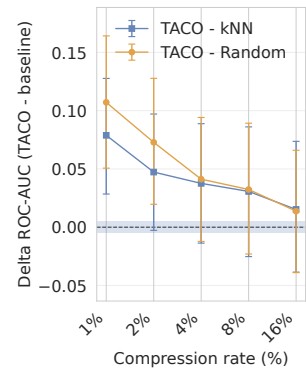

*Figure 8.* Comparison of TACO and the two compression baselines (kNN and Random). **Left:** ROC-AUC distributions across different compression rates. **Right:** Mean performance difference $\Delta$ROC-AUC (TACO $-$ baseline) across compression rates with **95% bootstrap confidence intervals**. Positive values indicate consistent performance improvements of TACO over the corresponding baseline.

*Table 3.* Results on MetroPT-3 (mean $\pm$ std over 5 seeds)

| Method | Precision | Recall | AUPRC |
|---|---|---|---|
| TACO (chunk) | **0.9886 $\pm$ 0.0089** | 0.7925 $\pm$ 0.0711 | 0.8955 $\pm$ 0.0084 |
| POT (Random) | 0.6362 $\pm$ 0.3529 | 0.6894 $\pm$ 0.0749 | 0.7293 $\pm$ 0.1118 |
| POT (kNN) | 0.4950 $\pm$ 0.4448 | 0.3169 $\pm$ 0.3049 | 0.3715 $\pm$ 0.3437 |
| TabPFNv2 (Random) | 0.6160 $\pm$ 0.4150 | 0.7333 $\pm$ 0.1095 | 0.7439 $\pm$ 0.1180 |
| TabPFNv2 (kNN) | 0.1867 $\pm$ 0.2520 | 0.3410 $\pm$ 0.3088 | 0.2358 $\pm$ 0.2036 |
| AutoGluon (extreme, 30m) | 0.9824 $\pm$ 0.0013 | **0.8886 $\pm$ 0.0032** | **0.9656 $\pm$ 0.0022** |
| XGBoost | 0.9722 $\pm$ 0.0015 | 0.8467 $\pm$ 0.0365 | 0.9593 $\pm$ 0.0072 |

**The results indicate that TACO consistently outperforms compression-related baselines across all the evaluation metrics, validating the chunk and stich approach.**

Moreover, to validate the generality of chunking, we run our method on all 36 TabArena classification datasets with 10 classes or fewer, including regimes where the predictor-only transformer is unable to operate due to the requirement of fitting the full dataset in memory. For more details, we refer the reader to Appendix B.

We additionally report results on the TabFSbench (Cheng et al., 2025) and TableShift (Gardner et al., 2024) benchmarks in the appendix in Section D, where we show that TACO can run successfully with competitive performance on tables with up to $\sim$ **6M rows**.

## 5. Limitations

Even though TACO advances the field of Tabular Foundation Models by achieving scalability to millions of training data points, it can still be improved along several dimensions. In particular, pretraining could benefit from more diverse priors that better capture real-world datasets, including those with a higher degree of missing values. Furthermore, incorporating datasets that reflect distribution shifts would be important for real-world deployment scenarios. Recent work (Helli et al., 2024) has shown that temporal distribution shift handling can be integrated into the TFM regime, and a similar approach could be applied to TACO. While we benchmark TACO on established tabular benchmarks following prior work (Qu et al., 2025; Hollmann et al., 2025;

Zhang et al., 2025), we acknowledge that our evaluation is currently limited to classification tasks. Extending to regression and time-series settings remains an important direction for future work.

## 6. Conclusion

Tabular Foundation Models have emerged as state-of-the-art for tabular classification. However, their inference cost scales with the conditional context. In this work, we tackle the quadratic complexity of transformer attention by introducing TACO, a novel modular architecture for fast context compression. TACO integrates a compressor module that learns a mapping from the input domain to a compact representation. A predictor module, then, is fed with the compressed data to produce test-point predictions. Our evaluation highlights that our model enables a 94x speedup and 98% memory consumption reduction without degrading performance significantly compared to the predictor-only transformer.

## Acknowledgments

We acknowledge funding by the Deutsche Forschungsgemeinschaft (DFG, German Research Foundation) under SFB 1597 (SmallData), grant number 499552394.

We also acknowledge the funding support from the "Bayerisches Landesamt für Steuer" for the Bavarian AI Taxation Laboratory.

Moreover, we gratefully acknowledge the scientific support and HPC resources provided by the Erlangen National High Performance Computing Center (NHR@FAU) of the Friedrich-Alexander-Universität Erlangen-Nürnberg (FAU) under the NHR project DeepSmall - Meta-learning for regularizing deep networks under small data regimes. NHR funding is provided by federal and Bavarian state authorities. NHR@FAU hardware is partially funded by the German Research Foundation (DFG) – 440719683.

## Impact Statement

This paper presents work whose goal is to advance the field of Machine Learning. There are many potential societal consequences of our work, none which we feel must be specifically highlighted here.

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

## A. Predict Time and Memory Consumption

### A.1. Time and Memory Consumption of TACO vs POT

In Insight 1, Figure 2 reports the combined fit-and-predict runtime on the synthetic grid when the test set is split into 10 batches. Here, we additionally report results for processing the full test set as a single batch in Figure 9, and for splitting it into 100 batches in Figure 10.

In Figure 9, the upper plot shows the setting without KV caching, and the lower plot shows the setting using KV caching. In both cases, the overall runtime is comparable. However, POT runs out of memory on 32 out of 100 datasets when KV Caching is used, reflecting the substantial memory required to fit the largest problem instances when the test set is processed as a single batch.

In contrast, Figure 10 reveals a marked shift. Without KV caching (upper plot), TACO dominates across the grid, as indicated by the blue color at essentially all grid points. POT, on the other hand, becomes prohibitively slow on the largest datasets, requiring up to $\sim 9$ hours to produce predictions. A similar pattern holds with KV caching enabled (lower plot): TACO remains consistently faster at fit + predict time and is more memory efficient, whereas POT again runs out of memory on 32 out of 100 datasets.

Moreover, we observe that even if POT was not constrained by memory, its inference time increases much more rapidly than that of TACO as the dataset size grows. This behavior is already evident at the 45K-row / 400-feature setting, where POT reaches the darkest red region, indicating prohibitively long runtimes. Additionally, we report in Figure 11, Figure 12, Figure 13, and Figure 14 separate heatmaps for fit and predict times for each test scenario.

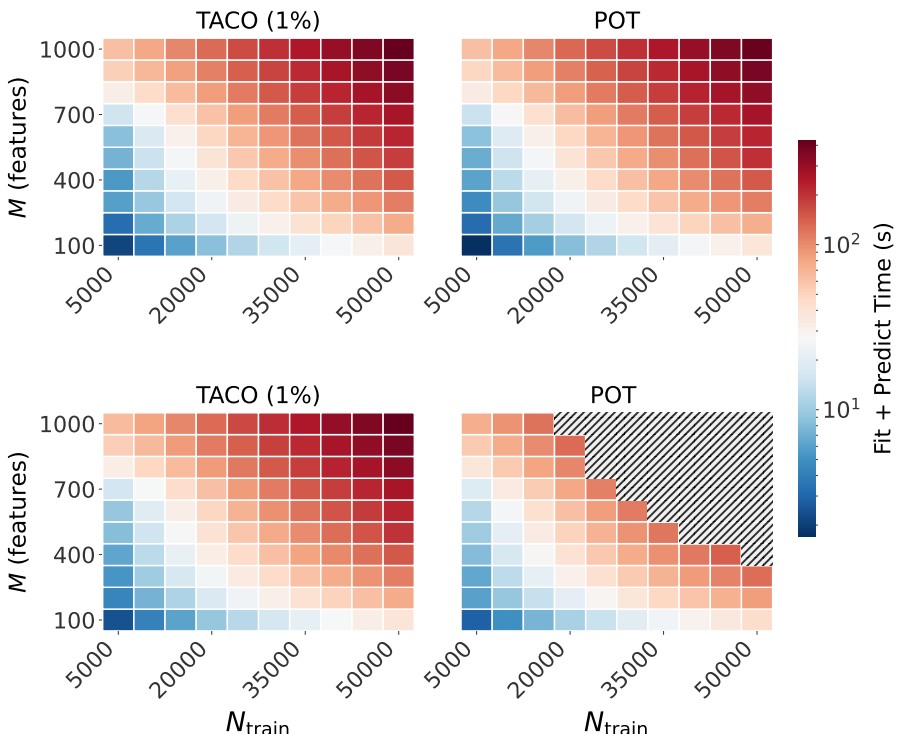

*Figure 9.* Fit+Predict Time Heatmap for the full test batch scenario. **Top:** Fit+Predict times for TACO and the Predictor Transformer, without using KV caching. **Bottom:** Predict times with KV caching. Black shaded regions indicate out-of-memory errors.

Lastly, we show empirically, in Table 4 that on a medium-scale dataset with $15,000$ rows and $500$ features, the predictor-only transformer requires $30.45\,\mathrm{GB}$ of GPU VRAM with KV caching. Compressing the context makes KV caching far more tractable: TACO (16%) reduces VRAM usage to $9.94\,\mathrm{GB}$, and TACO (1%) further reduces it to $8.69\,\mathrm{GB}$.

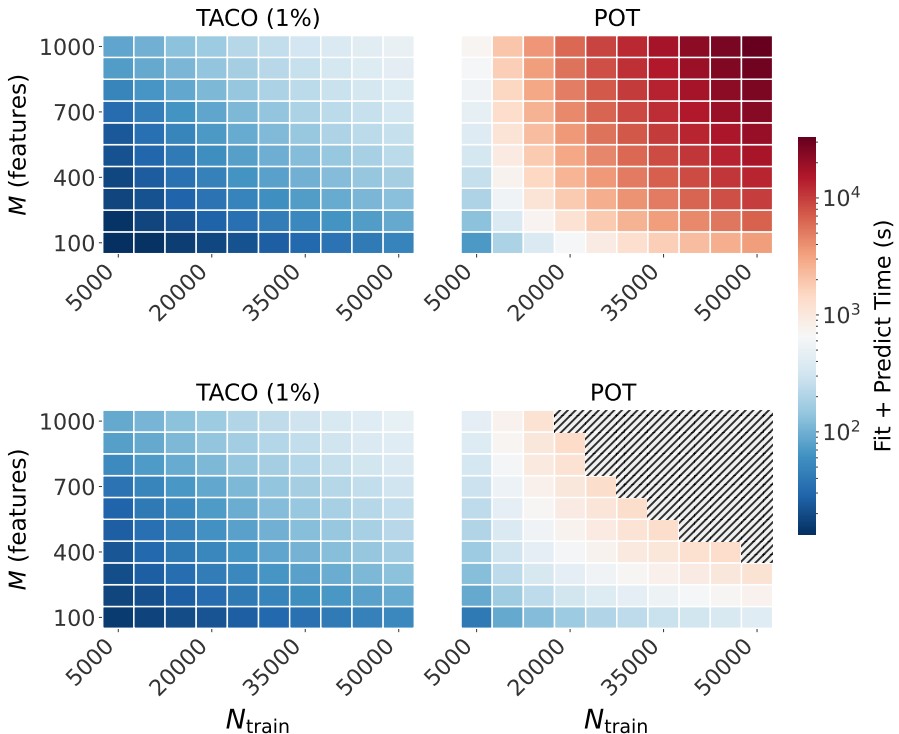

*Figure 10.* Fit+Predict Time Heatmap for the 100 test batches scenario. **Top:** Fit+Predict times for TACO and the Predictor Transformer, without using KV caching. **Bottom:** Predict times with KV caching. Black shaded regions indicate out-of-memory errors.

*Table 4.* Inference time and memory consumption for the predictor-only transformer (POT), versus TACO with different compression rates, with KV-caching.

| Method | Fit Time | First Predict Time | Subsequent Predict Time | Fit Mem | Predict Mem | |
|---|---|---|---|---|---|---|
| | | | Mean ± Std | | Mean | Median |
| POT | 40.60s | 7.30s | 7.14s±0.08s | 30.45GB | 8.24GB | 8.24GB |
| TACO 1% | 39.00s | 279ms | 198ms±4ms ($\times36.1$) | 8.69GB | 678MB ($-92\%$) | 678MB ($-92\%$) |
| TACO 2% | 39.59s | 305ms | 205ms±4ms ($\times34.8$) | 8.77GB | 975MB ($-88.4\%$) | 976MB ($-88.4\%$) |
| TACO 4% | 39.84s | 384ms | 217ms±6ms ($\times33$) | 8.94GB | 1.53GB ($-81.4\%$) | 1.53GB ($-81.4\%$) |
| TACO 8% | 41.65s | 487ms | 241ms±5ms ($\times29.6$) | 9.27GB | 2.69GB ($-67.3\%$) | 2.69GB ($-67.3\%$) |
| TACO 16% | 46.09s | 731ms | 283ms±7ms ($\times25.2$) | 9.94GB | 5.02GB ($-39.1\%$) | 5.02GB ($-39.1\%$) |

Parentheses: $\times$ speed-up, $-\%$ memory reduction (vs. POT).

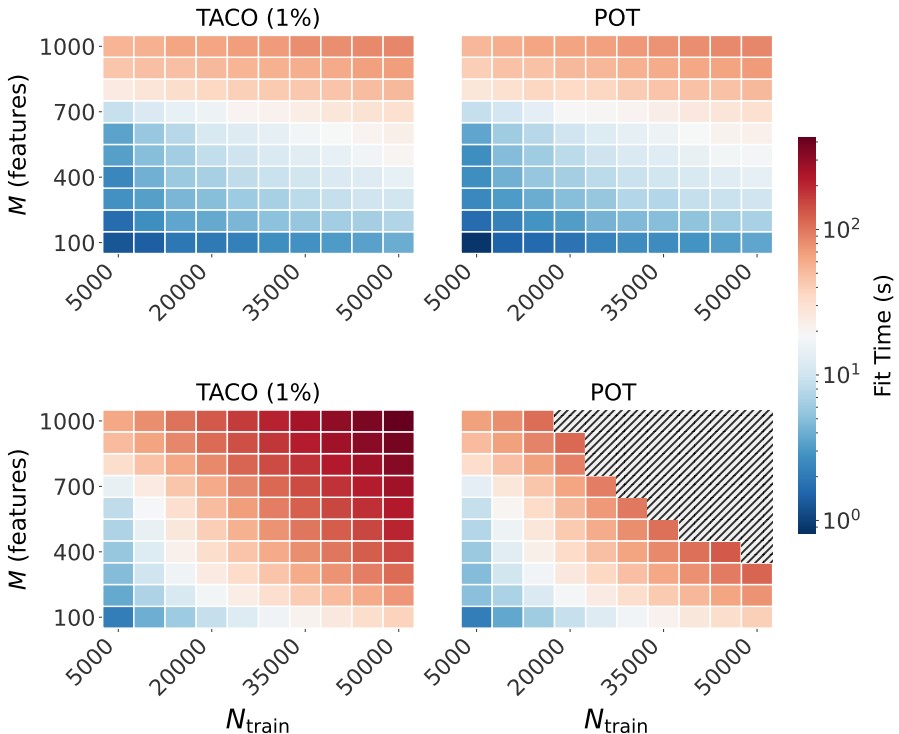

*Figure 11.* Fit Time Heatmap. **Top:** Fit times for TACO and the Predictor Transformer, without using KV caching. **Bottom:** Fit times with KV caching. Black shaded regions indicate out-of-memory errors.

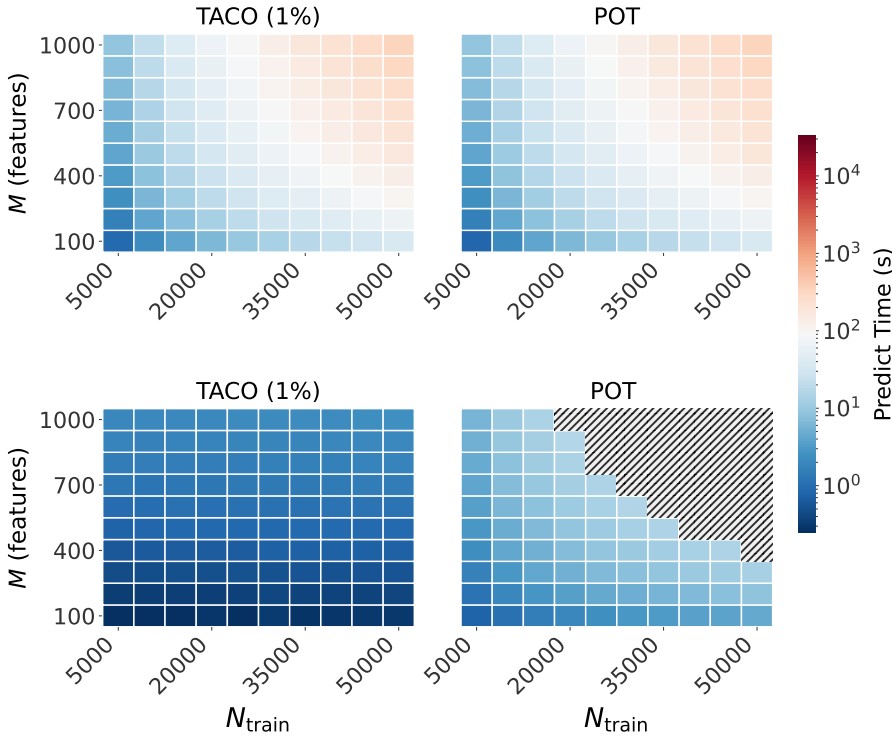

*Figure 12.* Predict time heatmap in the full test batch scenario. **Top:** Predict times for TACO and the Predictor Transformer, without using KV caching. **Bottom:** Predict times with KV caching. Black shaded regions indicate out-of-memory errors.

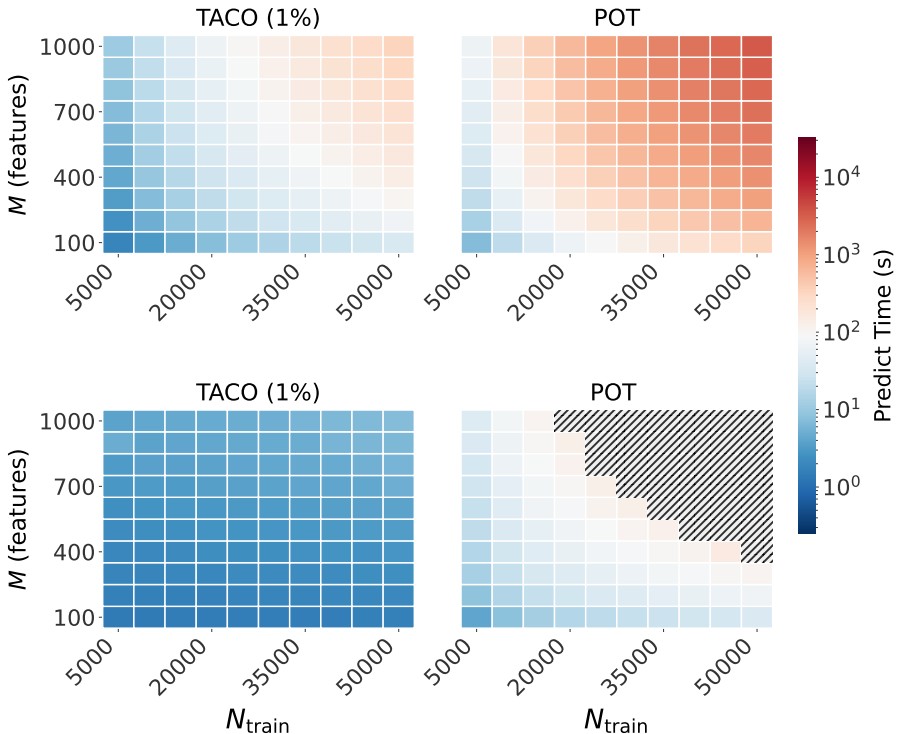

*Figure 13.* Predict time heatmap in the 10 test batches scenario. **Top:** Predict times for TACO and the Predictor Transformer, without using KV caching. **Bottom:** Predict times with KV caching. Black shaded regions indicate out-of-memory errors.

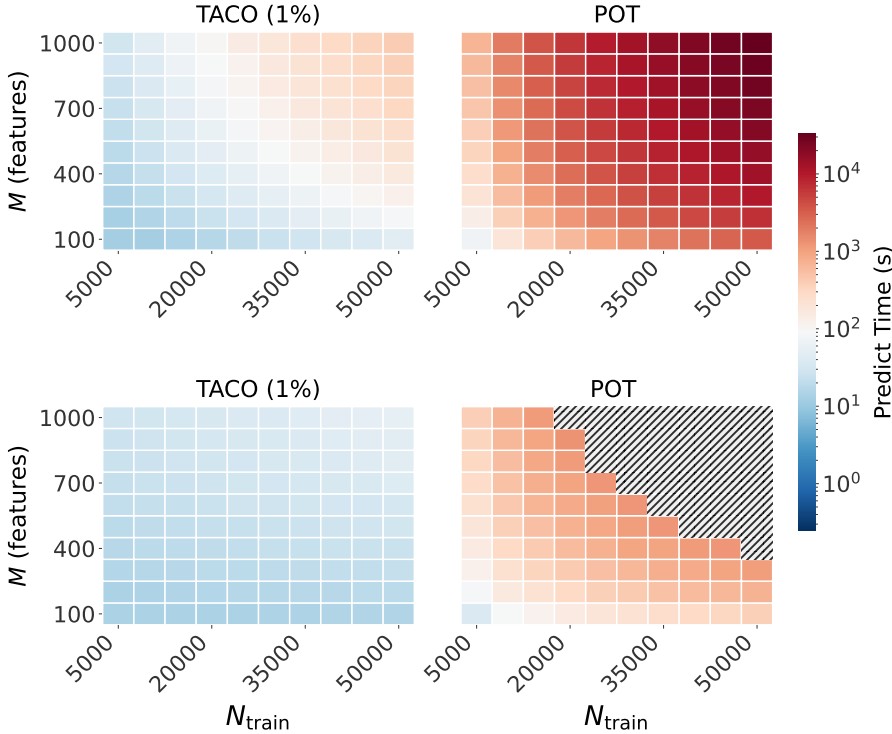

*Figure 14.* Predict time heatmap in the 100 test batches scenario. **Top:** Predict times for TACO and the Predictor Transformer, without using KV caching. **Bottom:** Predict times with KV caching. Black shaded regions indicate out-of-memory errors.

### A.2. Time and Memory Consumption of TACO vs compression baselines

Table 5 and Figure 15 report inference time and memory usage for TACO and the compression baselines for the same experimental setup presented in Table 4 in Section A.1. For TACO without KV-cache, inference consists of two phases: (i) a one-time context compression step performed during the first prediction, and (ii) subsequent predictions that reuse the compressed context. Since the compressed context is already embedded, subsequent predictions bypass the predictor's embedding layer and only execute the remaining inference stages.

As a result, the first prediction of TACO has a runtime comparable to the full-context predictor-only transformer and is therefore slower than inference with reduced-context variants, whereas subsequent predictions are substantially faster than those obtained even using the compression baselines.

At equal retained context ratios, TACO achieves between $1.2\times$ and $1.4\times$ lower mean prediction latency than kNN and random sampling, with the gap widening as the retained context fraction decreases (e.g., $306\,\mathrm{ms}$ vs. $408\,\mathrm{ms}$ at $1\%$). This advantage stems from the fact that kNN and random sampling remain stateless and must re-embed the selected context for every prediction, whereas TACO amortizes the compression cost across queries.

Compared to kNN and random sampling, TACO requires higher prediction-time memory at identical context ratios, as it retains a compressed *embedded* context to amortize embedding costs across predictions. However, this memory footprint remains negligible relative to the full predictor, yielding a $78$–$98\%$ reduction in prediction-time memory while supporting faster subsequent inference. Importantly, in contrast to kNN and random sampling, which exhibit pronounced performance degradation under context reduction, TACO maintains performance close to the full predictor as discussed in the main paper.

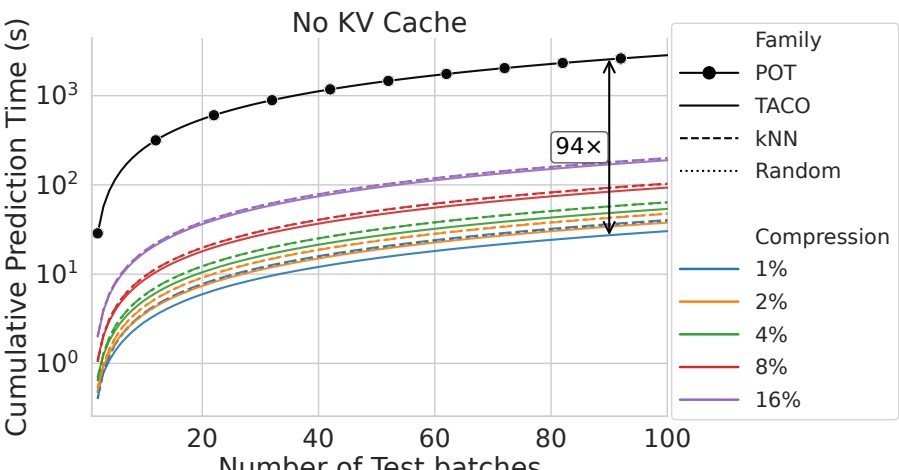

*Figure 15.* Cumulative prediction time for TACO at different compression rates $r$ compared to POT, random sampling and kNN for 100 test batches, without using KV Caching.

## B. TabArena Chunking Results

Since we use the TabPFNv2 architecture as our backbone for both the compressor and predictor, we first run TACO on the same TabArena subset as the original method. Results are presented in Table 1. Because TACO can employ chunking (Insight 6), we can then scale this evaluation to the full TabArena classification suite at an extreme compression rate of $0.1\%$ to probe how informative the compressor's latent context rows remain. We use a fixed compression rate of $0.1\%$ for TACO and the baselines across all datasets. While chunking and stitching is only necessary for large datasets, we nevertheless apply it to smaller datasets strictly to isolate and demonstrate its effectiveness. The TabArena classification datasets span a wide range in size, from $N = 748$ to $N = 150,000$ rows. We therefore adopt a simple chunk-size policy as a function of $N$:

$$C(N) = \begin{cases} 500, & N < 2,000, \\ 1,000, & 2,000 \leq N < 10,000, \\ 5,000, & 10,000 \leq N < 20,000, \\ 10,000, & N \geq 20,000. \end{cases}$$

*Table 5.* Inference Time and Memory Benchmark of TACO and Compression Baselines without KV-caching

| Method | Fit Time | First Predict Time | Subsequent Predict Time | Fit Mem | Predict Mem | |
|---|---|---|---|---|---|---|
| | | | Mean ± Std | | First Predict | Subsequent |
| Predictor | 8.41s | 28.55s | 28.67s±0.05s | 36MB | 22.45GB | 22.45GB |
| TACO 1% | 8.22s | 29.68s | 306ms±18ms ($\times 93.6$) | 92MB | 22.69GB | 549MB (−97.6%) |
| TACO 2% | 8.30s | 29.80s | 382ms±18ms ($\times 75.2$) | 92MB | 22.93GB | 845MB (−96.3%) |
| TACO 4% | 9.86s | 30.71s | 544ms±17ms ($\times 52.7$) | 92MB | 23.42GB | 1.41GB (−93.7%) |
| TACO 8% | 10.02s | 32.47s | 943ms±17ms ($\times 30.4$) | 92MB | 24.39GB | 2.56GB (−88.6%) |
| TACO 16% | 9.20s | 35.36s | 1.91s±0.01s ($\times 15$) | 92MB | 26.34GB | 4.89GB (−78.2%) |
| kNN 1% | 8.38s | 431ms | 408ms±5ms ($\times 70.3$) | 36MB | 395MB | 395MB (−98.3%) |
| kNN 2% | 7.95s | 495ms | 479ms±5ms ($\times 59.9$) | 36MB | 625MB | 625MB (−97.3%) |
| kNN 4% | 8.06s | 665ms | 645ms±5ms ($\times 44.5$) | 36MB | 1.06GB | 1.06GB (−95.3%) |
| kNN 8% | 8.57s | 1.08s | 1.05s±0.01s ($\times 27.4$) | 36MB | 1.95GB | 1.95GB (−91.3%) |
| kNN 16% | 7.96s | 2.04s | 2.02s±0.01s ($\times 14.2$) | 36MB | 3.74GB | 3.74GB (−83.4%) |
| Random 1% | 8.61s | 422ms | 398ms±6ms ($\times 72.1$) | 36MB | 396MB | 396MB (−98.3%) |
| Random 2% | 10.53s | 515ms | 483ms±6ms ($\times 59.4$) | 36MB | 625MB | 625MB (−97.3%) |
| Random 4% | 8.93s | 664ms | 643ms±6ms ($\times 44.6$) | 36MB | 1.06GB | 1.06GB (−95.3%) |
| Random 8% | 10.83s | 1.08s | 1.04s±0.01s ($\times 27.7$) | 36MB | 1.95GB | 1.95GB (−91.3%) |
| Random 16% | 8.83s | 2.05s | 2.01s±0.01s ($\times 14.3$) | 36MB | 3.74GB | 3.74GB (−83.4%) |

Parentheses: $\times$ speed-up, $-\%$ memory reduction (vs. POT).

With a compression rate $0.1\%$, this policy yields between 2 and 120 retained context rows per dataset after chunking and stitching, depending on $N$. We summarize performance across all datasets using boxplots in Figure 16. Overall, the results show that even at these extreme compression rates, the embeddings learned by the compressor remain highly informative, and TACO consistently outperforms both Random and kNN by a large margin.

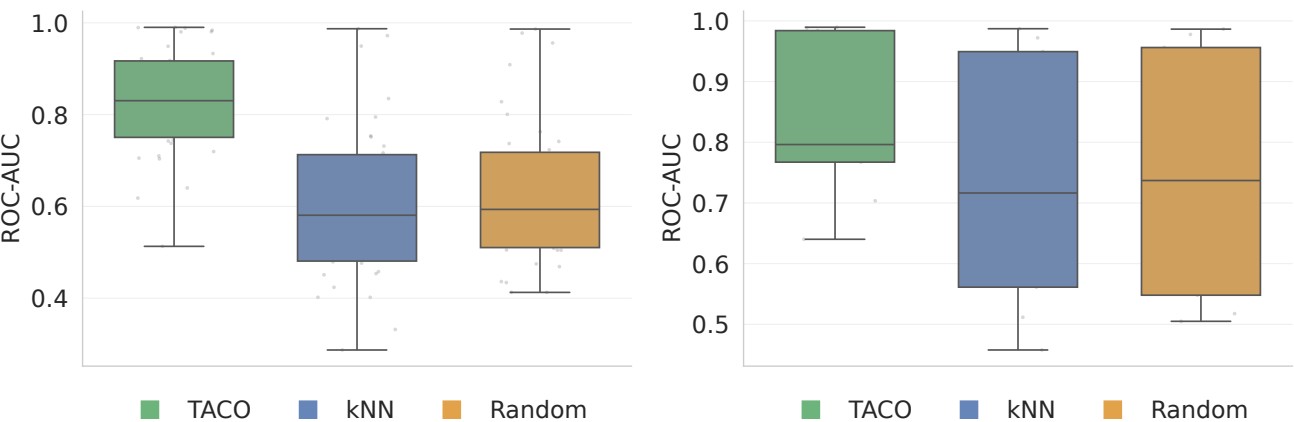

*Figure 16.* **Left:** Distribution of test performances on all TabArena classification tasks with chunking. **Right:** Distribution of test performances on TabArena classification tasks with $N \geq 20K$ rows, with chunking.

Table 6 reports the full results on all TabArena datasets. Here, $K$ denotes the number of latent context rows retained after compression, ranging from 2 for the smallest dataset to 120 for the largest. We show that, even with as few as $K = 2$ latent context rows, TACO achieves strong performance and outperforms the random sampling and kNN baselines by a large margin.

*Table 6.* Performance on the full TabArena suite (ROC-AUC; higher is better). K indicates number of latent context rows retained after compression.

| Dataset | #Rows | K | TACO | Random | kNN |
|---|---|---|---|---|---|
| blood-transfusion-service-center | 748 | 1 | **0.719** | 0.564 | 0.476 |
| diabetes | 768 | 2 | **0.817** | 0.436 | 0.453 |
| anneal | 898 | 2 | **0.980** | 0.412 | 0.588 |
| credit-g | 1000 | 2 | **0.742** | 0.475 | 0.486 |
| maternal_health_risk | 1014 | 2 | **0.856** | 0.632 | 0.553 |
| qsar-biodeg | 1054 | 2 | **0.922** | 0.434 | 0.402 |
| website_phishing | 1353 | 2 | **0.911** | 0.505 | 0.514 |
| Fitness_Club | 1500 | 2 | **0.805** | 0.413 | 0.287 |
| MIC | 1699 | 3 | **0.618** | 0.521 | 0.593 |
| Is-this-a-good-customer | 1723 | 3 | **0.710** | 0.557 | 0.451 |
| Marketing_Campaign | 2240 | 2 | **0.907** | 0.540 | 0.402 |
| hazelnut-spread-contaminant-detection | 2400 | 2 | **0.949** | 0.469 | 0.626 |
| seismic-bumps | 2584 | 2 | **0.794** | 0.515 | 0.424 |
| splice | 3190 | 3 | **0.990** | 0.602 | 0.561 |
| Bioresponse | 3751 | 3 | **0.867** | 0.615 | 0.508 |
| hiva_agnostic | 3845 | 3 | **0.513** | 0.509 | 0.479 |
| students_dropout_and_academic_success | 4424 | 3 | **0.844** | 0.505 | 0.615 |
| churn | 5000 | 4 | **0.910** | 0.577 | 0.614 |
| polish_companies_bankruptcy | 5910 | 4 | **0.918** | 0.646 | 0.332 |
| taiwanese_bankruptcy_prediction | 6819 | 5 | **0.933** | 0.723 | 0.835 |
| NATICUSdroid | 7491 | 5 | **0.981** | 0.909 | 0.731 |
| coil2000_insurance_policies | 9822 | 7 | **0.705** | 0.612 | 0.530 |
| Bank_Customer_Churn | 10000 | 7 | **0.855** | 0.763 | 0.751 |
| heloc | 10459 | 7 | **0.796** | 0.741 | 0.754 |
| jm1 | 10885 | 8 | **0.737** | 0.694 | 0.682 |
| E-CommereShippingData | 10999 | 8 | **0.745** | 0.586 | 0.658 |
| online_shoppers_intention | 12330 | 9 | **0.915** | 0.801 | 0.791 |
| in_vehicle_coupon_recommendation | 12684 | 9 | **0.768** | 0.601 | 0.574 |
| HR_Analytics_Job_Change_of_Data_Scientists | 19158 | 13 | **0.802** | 0.701 | 0.701 |
| credit_card_clients_default | 30000 | 20 | **0.788** | 0.737 | 0.716 |
| Amazon_employee_access | 32769 | 22 | **0.704** | 0.548 | 0.561 |
| bank-marketing | 45211 | 31 | **0.767** | 0.664 | 0.689 |
| kddcup09_appetency | 50000 | 34 | **0.796** | 0.505 | 0.458 |
| Diabetes130US | 71518 | 48 | **0.640** | 0.518 | 0.512 |
| APSFailure | 76000 | 51 | **0.990** | 0.978 | 0.972 |
| SDSS17 | 78053 | 53 | **0.988** | 0.986 | 0.987 |
| customer_satisfaction_in_airline | 129880 | 87 | **0.984** | 0.956 | 0.949 |
| GiveMeSomeCredit | 150000 | 100 | **0.858** | 0.828 | 0.795 |

## C. Detailed Results Across Datasets

This section presents additional details comparing TACO to the predictor-only transformer (POT) baseline across 26 classification datasets from TabArena (Erickson et al., 2025). Table 7 provides the full test ROC-AUC scores under different compression rates, along with the corresponding differences relative to POT. Across datasets, TACO yields performance that is very similar to POT, with only minor variations. This is also reflected in Figure 17, where the ROC-AUC distributions for POT and the TACO variants largely overlap.

*Table 7.* Test ROC-AUC performance of POT and TACO across datasets under varying compression rates (1%–16%). Parentheses report the absolute ROC-AUC difference of each TACO variant relative to POT (TACO - POT).

| Dataset | #Rows | POT | TACO (1%) | TACO (2%) | TACO (4%) | TACO (8%) | TACO (16%) |
|---|---|---|---|---|---|---|---|
| blood-transfusion-service-center | 748 | 0.717 | **0.741** (+0.024) | 0.740 (+0.023) | 0.715 (-0.002) | 0.711 (-0.006) | 0.702 (-0.015) |
| diabetes | 768 | 0.836 | 0.838 (+0.002) | 0.839 (+0.003) | 0.840 (+0.003) | **0.841** (+0.005) | 0.839 (+0.003) |
| anneal | 898 | **0.997** | 0.966 (-0.031) | 0.983 (-0.014) | 0.980 (-0.016) | 0.983 (-0.014) | 0.987 (-0.009) |
| credit-g | 1000 | 0.781 | 0.769 (-0.012) | 0.774 (-0.007) | 0.780 (-0.001) | **0.783** (+0.002) | **0.783** (+0.002) |
| maternal_health_risk | 1014 | **0.941** | 0.873 (-0.068) | 0.886 (-0.055) | 0.895 (-0.046) | 0.903 (-0.038) | 0.912 (-0.029) |
| qsar-biodeg | 1054 | **0.938** | 0.933 (-0.005) | 0.936 (-0.001) | 0.937 (-0.001) | 0.937 (-0.001) | 0.937 (-0.001) |
| website_phishing | 1353 | **0.970** | 0.953 (-0.017) | 0.958 (-0.012) | 0.962 (-0.008) | 0.963 (-0.007) | 0.963 (-0.007) |
| Fitness_Club | 1500 | 0.806 | **0.807** (+0.001) | 0.805 (-0.001) | 0.805 (-0.001) | 0.805 (-0.002) | 0.804 (-0.002) |
| MIC | 1699 | 0.663 | 0.666 (+0.003) | 0.668 (+0.005) | 0.670 (+0.006) | **0.671** (+0.007) | **0.671** (+0.008) |
| Is-this-a-good-customer | 1723 | 0.723 | 0.728 (+0.005) | **0.730** (+0.007) | **0.730** (+0.007) | 0.728 (+0.005) | 0.729 (+0.006) |
| Marketing_Campaign | 2240 | 0.931 | 0.930 (-0.001) | 0.932 (+0.001) | 0.933 (+0.002) | **0.934** (+0.003) | 0.933 (+0.002) |
| hazelnut-spread-contaminant-detection | 2400 | **0.977** | 0.972 (-0.005) | 0.973 (-0.004) | 0.974 (-0.003) | 0.975 (-0.002) | 0.975 (-0.002) |
| seismic-bumps | 2584 | 0.784 | **0.791** (+0.007) | 0.790 (+0.006) | 0.789 (+0.005) | 0.790 (+0.005) | 0.789 (+0.005) |
| splice | 3190 | **0.996** | **0.996** (-0.000) | **0.996** (+0.000) | **0.996** (+0.000) | **0.996** (+0.000) | **0.996** (+0.000) |
| students_dropout_and_academic_success | 4424 | 0.863 | 0.862 (-0.000) | 0.864 (+0.001) | 0.864 (+0.002) | **0.865** (+0.002) | **0.865** (+0.002) |
| churn | 5000 | 0.931 | **0.936** (+0.005) | **0.936** (+0.005) | 0.934 (+0.004) | 0.935 (+0.004) | 0.934 (+0.004) |
| polish_companies_bankruptcy | 5910 | **0.960** | 0.946 (-0.015) | 0.946 (-0.014) | 0.946 (-0.014) | 0.946 (-0.014) | 0.946 (-0.014) |
| taiwanese_bankruptcy_prediction | 6819 | 0.943 | 0.944 (+0.001) | **0.945** (+0.002) | **0.945** (+0.002) | **0.945** (+0.002) | **0.945** (+0.002) |
| NATICUSdroid | 7491 | **0.985** | 0.984 (-0.001) | 0.984 (-0.001) | 0.984 (-0.001) | 0.984 (-0.001) | 0.984 (-0.001) |
| coil2000_insurance_policies | 9822 | **0.751** | 0.749 (-0.001) | **0.751** (+0.000) | **0.751** (+0.001) | **0.751** (+0.001) | **0.751** (+0.000) |
| Bank_Customer_Churn | 10000 | **0.869** | 0.864 (-0.005) | 0.864 (-0.005) | 0.864 (-0.005) | 0.864 (-0.005) | 0.864 (-0.005) |
| heloc | 10459 | 0.799 | **0.800** (+0.001) | **0.800** (+0.001) | **0.800** (+0.001) | **0.800** (+0.001) | **0.800** (+0.001) |
| jm1 | 10885 | **0.775** | 0.745 (-0.029) | 0.746 (-0.028) | 0.746 (-0.028) | 0.747 (-0.028) | 0.746 (-0.028) |
| E-CommereShippingData | 10999 | 0.739 | **0.740** (+0.001) | 0.739 (+0.000) | 0.739 (+0.000) | 0.739 (+0.000) | 0.739 (-0.000) |
| online_shoppers_intention | 12330 | 0.923 | 0.924 (+0.001) | **0.925** (+0.001) | **0.925** (+0.002) | **0.925** (+0.002) | **0.925** (+0.002) |
| in_vehicle_coupon_recommendation | 12684 | **0.815** | 0.782 (-0.033) | 0.782 (-0.033) | 0.783 (-0.032) | 0.783 (-0.032) | 0.783 (-0.032) |

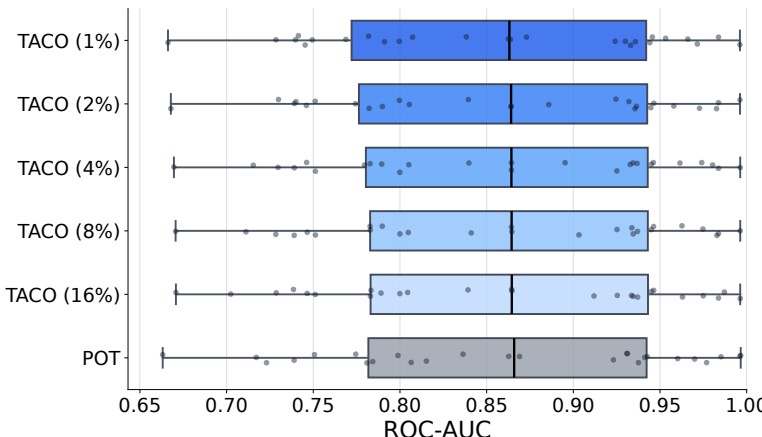

*Figure 17.* Boxplots of test ROC-AUC across datasets for the predictor-only transformer (POT) and TACO under varying compression rates (1%–16%). Boxes indicate the interquartile range with the median, while whiskers show the overall spread of performance.

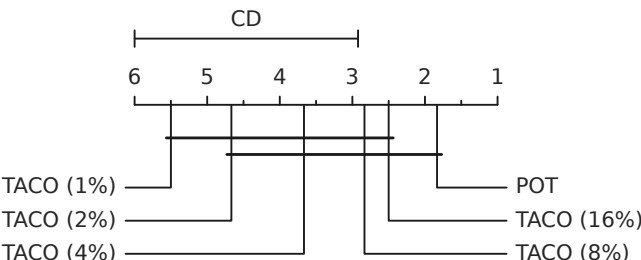

*Figure 18.* Critical difference (CD) diagram comparing predictor-only transformer (POT) and TACO on the `single` feature-shift regime of TabFSBench. Average ranks are computed using Friedman–Nemenyi tests (Demšar, 2006) via `autorank` (Herbold, 2020b) ($\alpha = 0.05$). Bars connect methods without significant differences.

## D. Additional Results on TabFSBench and TableShift

### D.1. Extended Evaluation on TabFSBench and TableShift

To further evaluate the robustness and scalability of compressed-context training beyond TabArena, we additionally evaluate TACO on the TabFSBench (Cheng et al., 2025) and TableShift (Gardner et al., 2024) benchmarks. These benchmarks provide complementary evaluation settings, including feature shifts, out-of-distribution evaluation protocols, and substantially larger datasets than those considered in the main benchmark suite.

**TabFSBench.** We first evaluate TACO under varying compression rates on the TabFSBench benchmark (Cheng et al., 2025). Figure 18 presents a critical difference (CD) diagram comparing POT against TACO across compression rates ranging from 1% to 16%. Despite operating in aggressively compressed training contexts, all TACO variants remain statistically competitive with POT across datasets. In particular, the 8% and 16% compression variants achieve average ranks comparable to or better than POT, suggesting that moderate compression can preserve predictive performance even under substantial reductions in effective context size.

**TableShift.** We additionally evaluate the final 91k-step checkpoints on the TableShift benchmark (Gardner et al., 2024), which explicitly studies robustness under feature and distribution shifts in open-environment tabular learning settings. While distribution-shift robustness is not the primary focus of this work, TableShift provides a useful complementary evaluation setting for understanding the behavior of compressed-context training beyond standard in-distribution benchmarks.

However, the benchmark exposes an important practical limitation of existing tabular foundation models. Out of the 15 TableShift datasets, 3 were inaccessible due to corrupted download links or access restrictions, and 10 caused POT and TabPFNv2 to run out of memory due to the large number of rows, leaving only 2 datasets (ANES and HELOC) where all methods could be evaluated directly. In contrast, TACO could be evaluated on all 12 accessible datasets. For the 2 compatible datasets, we used standard inference, while for the remaining 10 large-scale datasets, we used the chunk-and-stitch inference procedure enabled by compressed-context training.

The results are reported in Table 8. On the two datasets where all methods can be compared directly, compression does not degrade robustness under distribution shift. In particular, TACO achieves the strongest ID and OOD accuracy among all evaluated methods while remaining competitive on AUC metrics. On the remaining large-scale datasets, the chunked 0.1% compression variant achieves performance within approximately ±0.001 on average of XGBoost, despite operating in settings where POT and TabPFNv2 fail entirely due to memory constraints.

These results highlight a central contribution of TACO: compressed-context training enables tabular foundation models to scale to datasets that are otherwise infeasible for existing in-context tabular transformers. In the largest TableShift datasets, dataset sizes reach up to several million rows, yet TACO remains operational and competitive with strong classical baselines while standard TFMs cannot be executed due to memory limitations. For comparisons against additional non-TFM baselines, we refer the reader to Tables 4–11 of the original TableShift paper.

*Table 8.* TableShift results under in-distribution (ID) and out-of-distribution (OOD) evaluation settings. "OOM" denotes out-of-memory failures.

| Dataset | TabPFNv2 | POT | TACO 1% | TACO 2% | TACO 4% | TACO 8% | TACO 16% | TACO chunking 0.1% |
|---|---|---|---|---|---|---|---|---|
| **ID Accuracy** | | | | | | | | |
| ANES | 0.8648 | 0.8673 | 0.8597 | 0.8571 | 0.8584 | 0.8610 | 0.8610 | - |
| HELOC | 0.7446 | 0.7374 | 0.7482 | 0.7482 | 0.7482 | 0.7482 | 0.7482 | - |
| ASSISTments | OOM | OOM | - | - | - | - | - | 0.9306 |
| College Scorecard | OOM | OOM | - | - | - | - | - | 0.9353 |
| Hospital Readmission | OOM | OOM | - | - | - | - | - | 0.6424 |
| Diabetes | OOM | OOM | - | - | - | - | - | 0.8759 |
| Food Stamps | OOM | OOM | - | - | - | - | - | 0.8343 |
| Unemployment | OOM | OOM | - | - | - | - | - | 0.9719 |
| Income | OOM | OOM | - | - | - | - | - | 0.8065 |
| Public Health Insurance | OOM | OOM | - | - | - | - | - | 0.7846 |
| Sepsis | OOM | OOM | - | - | - | - | - | 0.9881 |
| Hypertension | OOM | OOM | - | - | - | - | - | 0.6587 |
| **ID AUC** | | | | | | | | |
| ANES | 0.8815 | 0.8747 | 0.8734 | 0.8738 | 0.8736 | 0.8738 | 0.8742 | - |
| HELOC | 0.7140 | 0.7119 | 0.7033 | 0.7107 | 0.7078 | 0.7084 | 0.7049 | - |
| ASSISTments | OOM | OOM | - | - | - | - | - | 0.9626 |
| College Scorecard | OOM | OOM | - | - | - | - | - | 0.9682 |
| Hospital Readmission | OOM | OOM | - | - | - | - | - | 0.6751 |
| Diabetes | OOM | OOM | - | - | - | - | - | 0.8143 |
| Food Stamps | OOM | OOM | - | - | - | - | - | 0.8227 |
| Unemployment | OOM | OOM | - | - | - | - | - | 0.9694 |
| Income | OOM | OOM | - | - | - | - | - | 0.8760 |
| Public Health Insurance | OOM | OOM | - | - | - | - | - | 0.7373 |
| Sepsis | OOM | OOM | - | - | - | - | - | 0.6802 |
| Hypertension | OOM | OOM | - | - | - | - | - | 0.7014 |
| **OOD Accuracy** | | | | | | | | |
| ANES | 0.8485 | 0.8460 | 0.8462 | 0.8460 | 0.8450 | 0.8450 | 0.8448 | - |
| HELOC | 0.4398 | 0.4310 | 0.4406 | 0.4436 | 0.4562 | 0.4552 | 0.4578 | - |
| ASSISTments | OOM | OOM | - | - | - | - | - | 0.5834 |
| College Scorecard | OOM | OOM | - | - | - | - | - | 0.8232 |
| Hospital Readmission | OOM | OOM | - | - | - | - | - | 0.5992 |
| Diabetes | OOM | OOM | - | - | - | - | - | 0.8314 |
| Food Stamps | OOM | OOM | - | - | - | - | - | 0.8006 |
| Unemployment | OOM | OOM | - | - | - | - | - | 0.9603 |
| Income | OOM | OOM | - | - | - | - | - | 0.7884 |
| Public Health Insurance | OOM | OOM | - | - | - | - | - | 0.5596 |
| Sepsis | OOM | OOM | - | - | - | - | - | 0.9245 |
| Hypertension | OOM | OOM | - | - | - | - | - | 0.5779 |
| **OOD AUC** | | | | | | | | |
| ANES | 0.8862 | 0.8821 | 0.8832 | 0.8830 | 0.8831 | 0.8830 | 0.8830 | - |
| HELOC | 0.7239 | 0.6935 | 0.7026 | 0.7012 | 0.7034 | 0.6994 | 0.7005 | - |
| ASSISTments | OOM | OOM | - | - | - | - | - | 0.6426 |
| College Scorecard | OOM | OOM | - | - | - | - | - | 0.8680 |
| Hospital Readmission | OOM | OOM | - | - | - | - | - | 0.6564 |
| Diabetes | OOM | OOM | - | - | - | - | - | 0.8044 |
| Food Stamps | OOM | OOM | - | - | - | - | - | 0.8002 |
| Unemployment | OOM | OOM | - | - | - | - | - | 0.9634 |
| Income | OOM | OOM | - | - | - | - | - | 0.8716 |
| Public Health Insurance | OOM | OOM | - | - | - | - | - | 0.7087 |
| Sepsis | OOM | OOM | - | - | - | - | - | 0.6329 |
| Hypertension | OOM | OOM | - | - | - | - | - | 0.6763 |

