# OpenReview forum: "End-to-End Compression for Tabular Foundation Models"
_ICML.cc/2026/Conference — ICML 2026 spotlight_

### Official Review · Reviewer_9YHr · 2026-03-02

**Soundness:** 2
**Presentation:** 1
**Significance:** 3
**Originality:** 3
**Overall Recommendation:** 4
**Confidence:** 4

**Summary:**

This paper introduces TACO, a learned, end-to-end compression framework for tabular in-context learning that reduces the effective training context from N rows to K≪N rows before prediction. The system comprises a compressor transformer that writes the information of the full training table into K rows and a predictor transformer that conditions on this compressed context to make predictions for test queries.

**Compliance With Llm Reviewing Policy:**

Affirmed.

**Final Justification:**

I acknowledge to see that the author has added limitations and conducted experiments in open environments. For this reason, I have raised the score to 4.

**Key Questions For Authors:**

1. Why were the official TabArena Elo metrics not used for performance comparison?

2. Why were the final results of TACO not submitted to the TabArena leaderboard?

3. Why were the model's performance not evaluated on regression tasks, missing data scenarios, or dynamic environments (TableShift, TabFSBench, and CLIMB)? I believe it is also crucial to validate the model's performance under such settings.

- Gardner, Josh, Zoran Popovic, and Ludwig Schmidt. "Benchmarking distribution shift in tabular data with tableshift." Advances in Neural Information Processing Systems 36 (2023): 53385-53432.
- Cheng, Zi-Jian, et al. "TabFSBench: Tabular Benchmark for Feature Shifts in Open Environments." arXiv preprint arXiv:2501.18935 (2025).
- Liu, Zhining, et al. "CLIMB: Class-imbalanced learning benchmark on tabular data." arXiv preprint arXiv:2505.17451 (2025).

**Limitations:**

This paper did not discuss any limitation of TACO, resulting in an incomplete article structure. However, I believe that TACO still has several limitations, such as whether the claim of "Hypothesis 1: TACO achieves up to 53× speedup during inference, without significant degradation in performance." has been genuinely validated by TabArena.

**Strengths And Weaknesses:**

Strengths：

1. Proposes a simple, modular, end-to-end learnable compressor for tabular foundation models that explicitly targets the quadratic attention bottleneck by operating on the data context rather than architectural attention approximations.

2. Clear high-level motivation and system description with a coherent training/evaluation protocol and sensible hypotheses structuring.

Weakness：

1. Incomplete paper structure, lacking a Limitation section (poor presentation).

2. Insufficiently rigorous experimental setup: only the TabArena dataset was used, without reference to TabArena's evaluation metrics, and the model results were not uploaded to TabArena for public disclosure.

3. Limited experimental scope: lacking evaluation on regression tasks, scenarios with missing values, and dynamic environments (fair soundness).

Although TACO novel, the approach merely claims effectiveness without official validation (obtaining official validation does not involve any violation of anonymity), and the experimental scope is too limited to demonstrate the robustness of TACO. If the authors can address the key questions below, I will raise my rating.

---

> ### Author Rebuttal · Authors · 2026-03-30
>
> We thank the reviewer for the feedback. Below we address the main concerns from the reviewer:
>
> ## Regarding regression results
> TACO's compressor takes the full training table as input and compresses it into a smaller set of latent rows, this process is agnostic to whether the target variable is categorical or continuous, as the compression mechanism treats all columns identically. However, demonstrating this empirically requires a backbone that was pretrained for regression, which in the case of TabPFN v2 is a separately trained model. This amounts to a full pretraining run (approximately 18 days on 8×H100 GPUs), which is infeasible within the rebuttal period. We would kindly point out to the reviewer that the current accepted open-source state-of-the-art work [1], only provides a classification model.
>
> ## Regarding the limitations section
> Because of the limited number of pages, we decided to provide more insights for the readers and opted out of explicitely writing a limitations section.
> We will explicitely list the limitations for the camera-ready version of our work.
>
> ## Regarding the poor presentation and incomplete structure:
> While, we agree we have not explicitly listed the limitations, listing limitations is not a mandatory requirement:
>
> https://media.icml.cc/Conferences/ICML2026/Styles/example_paper.pdf
>
> From our perspective, we do not believe that not listing limitations equates to a work with poor presentation and incomplete structure.
>
> Additionally, all fellow reviewers wrote the following about our work:
> - Reviewer Wdoc: “Presentation is clean, very easy to understand”
> - Reviewer eA7L:  “The writing is clear”
> - Reviewer jHfk: “The paper is well written and clearly structured”
>
> Could the reviewer provide a few reasons why our work is poorly presented and incomplete in contrary to what the other reviewer’s list?
>
> ## Regarding the TabArena Elo Metrics
>
> We would like to kindly point out to the reviewer that the ROC-AUC is also an official TabArena metric [2], our evaluation was conducted using TabArena's datasets, preprocessing pipeline, and same cross-validation fold, so the results are fully compatible with the benchmark. Additionally we would like to remind the reviewer that the ELO metric is derived from ROC-AUC.
>
> We have evaluated both TACO and POT on TabArena using the official evaluation pipeline and default metrics, including Elo.  This is the leaderboard state as of the moment of writing the reply:
>
> https://anonymous.4open.science/r/taco-AAA1/rebuttal/full_tabarena_elo_results.png
>
> ## Regarding the TabArena Leaderboard Submission
> We would like to point out to the reviewer that we have already contacted the authors of TabArena, and official submissions must be open-source  [2] (Appendix E.2)   which would break double-blind anonymity during the review period. We have instead evaluated locally using TabArena's official pipeline, and we have open-sourced our full codebase and model checkpoints in the anonymous repository. The reviewer is welcome to verify the results by pulling the leaderboard state locally.
>
> ## Regarding Missing Data:
> We would like to point out to the reviewer that TabArena already includes datasets with missing values: MIC (3.45% missing), polish_companies_bankruptcy (1.23% missing), Fitness_Club, Marketing_Campaign, and jm1. TACO handles these without any special treatment, as the compression module inherits the missing data handling of the underlying TFM.
>
> ## Regarding additional benchmarks:
> We thank the reviewer for the valuable suggestions. We ran on the TabFS benchmark suggested from the reviewer and provide the following results:
>
> https://anonymous.4open.science/r/taco-AAA1/rebuttal/TabFS_POT_vs_TACO_CD_diagram.png
>
> As observed, the results are consistent and TACO has no significant degradation in performance compared to POT.
>
> We believe to have clarified in detail all the concerns listed by the reviewer, based on the reviewer’s promise, we would kindly invite the reviewer to increase the rating and recommend acceptance.
>
>
> [1] Qu, J., Holzmüller, D., Varoquaux, G., & Le Morvan, M. (2025, October). TabICL: A Tabular Foundation Model for In-Context Learning on Large Data. In International Conference on Machine Learning (pp. 50817-50847). PMLR.
>
> [2] Erickson, N., Purucker, L., Tschalzev, A., Holzmüller, D., Desai, P. M., Salinas, D., & Hutter, F. (2025, December). TabArena: A Living Benchmark for Machine Learning on Tabular Data. In NeurIPS 2025-The Thirty-ninth Annual Conference on Neural Information Processing Systems Datasets and Benchmarks Track.

---

> > ### Author Rebuttal · Reviewer_9YHr · 2026-04-04
> >
> > Thanks to the authors' responses. I acknowledge the authors' explanation regarding the inability to add regression tasks. I still hold some concerns.
> >
> > 1. I believe limitations are an indispensable part of a paper. If, as the authors claim, limitations are not mandatory requirements, then why do ICML specifically highlight limitations as a separate section when we write reviews? Moreover, Reviewer eA7L and jHkF both mentioned this. From my perspective, we do believe that not listing limitations equates to a work with poor presentation and incomplete structure.
> >
> > 2. The authors' supplementary experiments in open environments only evaluated TabFS but did not assess more crucial challenges such as distribution shifts (TableShift), and no explanation was provided for this omission.

---

> > > ### Author Response · Authors · 2026-04-04
> > >
> > > We thank the reviewer for the reply and for the continued engagement.
> > >
> > > - We respectfully see the lack of explicit limitations equating to poor presentation and incomplete structure differently and we believe the other reviewers share our opinion:
> > >    - **Reviewer Wdoc:** “Presentation is clean, very easy to understand”
> > >    - **Reviewer eA7L:** “The writing is clear”
> > >    - **Reviewer jHfk:** “The paper is well written and clearly structured”
> > >
> > >    **Edit: 07.04.2026**
> > >
> > >    That said, we fully agree that explicitly discussing limitations is important. To address this, we provide a draft below and will include a dedicated limitations section in the camera-ready version:
> > >
> > >    **Even though TACO advances the field of Tabular Foundation Models by achieving scalability to millions of training data points, it can still be improved along several dimensions. In particular, pretraining could benefit from more diverse priors that better capture real-world datasets, including those with a higher degree of missing values. Furthermore, incorporating datasets that reflect distribution shifts would be important for real-world deployment scenarios, recent work [1] has shown that temporal distribution shift handling can be integrated into the TFM regime, and a similar approach could be applied to TACO. While we benchmark TACO on established tabular benchmarks following prior work [2,3,4], we acknowledge that our evaluation is currently limited to classification tasks. Extending to regression and time-series settings remains an important direction for future work.**
> > >
> > >    **Edit: Complete TableShift Results**
> > >
> > > - We thank the reviewer for raising the question regarding distribution shift scenarios. We already included TabFS following the reviewer's suggestion, a benchmark with feature shifts in tabular data in open-environment scenarios. While we agree that additional distribution shift benchmarks are important, they fall outside the primary scope of our work. Additionally, the proposed benchmark is not fully compatible with the experimental protocol, as the considered methods are constrained in the dataset sizes they can handle [2][3][4]. Out of 15 datasets, 3 are inaccessible due to corrupt download links or access restrictions, and 10 cause POT and TabPFNv2 to run out of memory due to the large number of rows, leaving only 2 (HELOC and Voting) where all methods can be compared. We ran TACO on all 12 accessible datasets, using standard inference on the 2 compatible datasets and chunk-and-stitch on the remaining 10, while POT and TabPFNv2 could only run on the 2 compatible datasets. We report all results below:
> > >
> > >    https://anonymous.4open.science/r/taco-AAA1/rebuttal/tableshift.md
> > >
> > >    On the 2 datasets where all methods can be compared, compression does not hurt performance under distribution shift, TACO achieves the highest ID and OOD accuracy among all methods, outperforming both POT and TabPFNv2, while remaining competitive on AUC. On the 10 larger datasets, TACO performs within ±0.001 on average of XGBoost, demonstrating that it can operate competitively in regimes where other TFMs cannot. These results highlight a core contribution of our work: TACO is the only method that enables TFMs to operate on these large-scale datasets, where datasets go up to 6M rows, achieving competitive performance with established baselines while existing TFMs fail due to memory constraints. For comparison with other baselines, we refer the reviewer to Tables 4–11 in the TableShift paper [5].
> > >
> > > **We believe to have addressed all the concerns from the reviewer, as such, we would like to invite the reviewer, according to his\her promise to increase the score and recommend acceptance following the clarifications.**
> > >
> > > [1] Helli, K., Schnurr, D., Hollmann, N., Müller, S., & Hutter, F. (2024). Drift-Resilient TabPFN: In-Context Learning Temporal Distribution Shifts on Tabular Data. NeurIPS.
> > >
> > > [2] Qu, J., Holzmüller, D., Varoquaux, G., & Le Morvan, M. (2025, October). TabICL: A Tabular Foundation Model for In-Context Learning on Large Data. In International Conference on Machine Learning (pp. 50817-50847). PMLR.
> > >
> > > [3] Hollmann, N., Müller, S., Purucker, L., Krishnakumar, A., Körfer, M., Hoo, S. B., ... & Hutter, F. (2025). Accurate predictions on small data with a tabular foundation model. Nature, 637(8045), 319-326.
> > >
> > > [4] Zhang, X., Maddix, D. C., Yin, J., Erickson, N., Ansari, A. F., Han, B., ... & Wang, B. Mitra: Mixed Synthetic Priors for Enhancing Tabular Foundation Models. In The Thirty-ninth Annual Conference on Neural Information Processing Systems.
> > >
> > > [5] Gardner, Josh, Zoran Popovic, and Ludwig Schmidt. "Benchmarking distribution shift in tabular data with tableshift." Advances in Neural Information Processing Systems 36 (2023): 53385-53432.

---

### Official Review · Reviewer_jHkF · 2026-03-04

**Soundness:** 4
**Presentation:** 4
**Significance:** 3
**Originality:** 3
**Overall Recommendation:** 5
**Confidence:** 4

**Summary:**

The paper introduces TACO, a tabular foundation model architecture/method that compresses training samples into much fewer tokens before including the test samples, leading to faster inference times while preserving good accuracy. This can also be used to process large datasets in chunks. The authors provide some ablations that lead to insights on runtime, memory, and different pre-training options.

**Compliance With Llm Reviewing Policy:**

Affirmed.

**Final Justification:**

The paper introduces a clean concept for training a model that compresses a dataset before including data, therefore facilitating higher inference efficiency. While the concept is not validated at state-of-the-art performance (lack of a state-of-the-art open-source prior + lack of large-dataset post-training) or for regression, it is plausible that simply adding these features would lead to a strong model with much better inference time and memory requirements. This is due to the extensive ablations of the proposed improvements and the comparison to a baseline trained under the same settings. The authors have addressed my concerns in the rebuttal. Therefore, I recommend acceptance.

**Key Questions For Authors:**

(Q1) l. 318 right: why does TACO have a higher fit-time memory footprint? At least for pre-training that's non-negligible because it limits the size of datasets you can pre-train on.
Also: why is there onlx 3x memory reduction in the KV caching setting, and why does POT use less memory with KV caching than without?

**Limitations:**

Many limitations are somewhat evident from reading the paper, but the authors do not provide a dedicated discussion of limitations.

**Strengths And Weaknesses:**

**Strengths**:
- Significance: The problem is relevant and timely, as tabular foundation models are popular but are costly at inference. The method slightly degrades performance, but provides strong benefits to inference time and memory usage.
- Originality: The method is new. It is also simple and easy to adapt, which I appreciate.
- Presentation: The paper is well-written and clearly structured.
- Open-source code is provided.

**Weaknesses**:

(W1) Evaluation: I like many parts of the evaluation, including the ablations. However, while Table 1 compares to TabICL and TabPFNv2, they have identical performance, and it is hard for me to compare the gap between POT and TACO to gaps between other ML methods since no other ML methods are included. It would be nice to see some more baselines to judge whether the model from the paper is competitive with some other baselines. If the authors don't use TabArena to evaluate their models that might be a bit more challenging, but at least a default random forest should be easy to do, and for boosted trees some cross-validated evaluation using AutoGluon or pytabkit or so could also be reasonably easy to do.

**Comments** (not necessary to address all of them in the rebuttal):
- l. 146: output dimension should probably not always be 1
- l. 156: are you calling it a decoder because test only attends to train? I would have said it's more of an encoder?
- l. 293 right: "POT and TACO have not been trained for as long as competitor methods"
- l. 327: I don't love the conclusion based on lack of statistical significance - the fact that the difference is not significant depends on the choice of test, inclusion of Bonferroni correction, inclusion of other baselines, and the number of datasets (which is on the small side since TabArena doesn't have that many datasets). Generally you can reject null hypotheses but you usually can't accept them (although the text is not exactly framed that way).
- Some hypotheses (e.g., 1 and 2) have a very detailed formulation that one wouldn't expect if these were actual hypotheses conceived before testing them. An idea would be to call them "insights" instead.
- The differences between TACO (1%) and TACO (16%) seem to be often much smaller than the differences between TACO (16%) and POT. I wonder if there is something that could be learned from training TACO (100%) - maybe there are two different bottlenecks: compression and something else. That something else could be the difficulty of copying from D^train to D^mini-train or the fact that the predictor shares weights between D^mini-train and x^test even though one is much more preprocessed than the other.
- I wonder a bit if the pre-training on only small dataset sizes has an influence on the comparison. But the predictor module is also only pre-trained on smaller sizes than what it sees at test-time, so maybe it balances out.

---

> ### Author Rebuttal · Authors · 2026-03-30
>
> We thank the reviewer for the valuable feedback. Below we address the major points from the reviewer.
>
> ## Regarding Baselines
> We agree that additional baselines help situate TACO in the broader landscape. Given that we run on TabArena, we pulled the leaderboard state with both POT and TACO present at the moment of writing this response.
> We present the results in the following figure:
>
> https://anonymous.4open.science/r/taco-AAA1/rebuttal/full_tabarena_elo_results.png
>
> ## Regarding Memory Footprint
> We thank the reviewer for raising interesting questions.
>
> **Why does TACO have higher fit-time memory compared to POT?** This is only the case for the setting without KV caching. During fit, TACO loads both the compressor and predictor onto the GPU, resulting in a higher fit-time memory. However, this can be further optimized, and the compressor can be removed from memory after the dataset is compressed. We would kindly point the reviewer to Table 2 (Line 330), where as observed, fit memory has a smaller impact compared to predict memory in terms of absolute size. As such, the larger fit memory consumption that TACO faces has a negligible impact on the overall memory consumption.
>
> **Why is there only a 3x memory reduction with KV caching than without?**
> We would like to point the reviewer to Section 3.2 (Line 141 right), where we explain in detail the reduction in memory between POT and TACO with and without KV caching. Summarized, without KV caching the reduction in complexity is (N_train/ K) ^ 2 , while with KV caching it is (N_train / K) where K is the number of rows after compression.
>
> **Why does POT use less memory with KV caching than without?** Without KV caching, every predict call recomputes attention over all N training rows plus the test rows from scratch, which is where the 22.45GB predict memory comes from. With KV caching, the key-value representations for the training rows are precomputed during fit, this is why fit memory increases from 36MB to 30.45GB. During prediction, the model only computes attention for the test rows against the cached KVs, reducing the prediction memory to 8.24GB. In other words, the memory cost shifts from predict to fit.
>
> ## Minor comments
>
> - *l. 146 (output dimension):* The reviewer is correct,  the output space should be ℝ^C (as in line 136), where C is the number of classes, rather than ℝ, given that we focus on classification. We will correct this in the revision.
>
> - *l. 156 (decoder vs encoder):* The reviewer raises a fair point. The architecture is bidirectional among training rows but applies causal masking so that test points attend to training points without attending to each other, which motivated our use of "decoder." We acknowledge that the terminology can be confusing and will clarify this in the revision.
>
> - *l. 293 (training length):* We want to clarify the training setup. POT and TACO use the TabPFNv2 architecture, but we do not have access to the TabPFNv2 prior and instead use the TabICL prior. Following TabICL's training procedure, we trained both TACO and POT on ~80M datasets over 80k steps, following stage 1 from TabICL's training pipeline. Furthermore, following suggestions from reviewer eA7L, we continued finetuning both TACO and POT on real data following an approach inspired by RealTabPFN and TabDPT, reaching 91K steps, amounting to ~91M datasets.  This is still considerably less than the 130M datasets used by TabPFNv2. The results for the TabArena ELO evaluation and the results posted in the reply to eA7L reflect the performance after the extra training steps.
>
> - *l. 327 (statistical significance):* We note that our phrasing, "no significant degradation", reports the outcome of the statistical test rather than claiming equivalence, which the reviewer also acknowledges. We additionally note that our updated evaluation on TabArena with Elo rankings, including other baselines, shows TACO and POT performing neck and neck, further supporting this result.
>
>    https://anonymous.4open.science/r/taco-AAA1/rebuttal/full_tabarena_elo_results.png
>
> - *Hypotheses naming:* We will revise the hypothesis naming following the reviewer’s suggestion.
>
> - *TACO(1%) vs TACO(16%) vs POT gap:* It is a valuable suggestion, we will pursue it for the camera-ready version of our work.
>
> - *Pretraining on small dataset sizes:* The reviewer's intuition is correct, both TACO and POT are pretrained on the same context sizes, so any out-of-distribution effects at test time apply equally to both.
>
> We believe to have addressed all the concerns raised by the reviewer. If the reviewer has additional concerns, we are happy to answer them.

---

> > ### Author Rebuttal · Reviewer_jHkF · 2026-04-02
> >
> > My comment about weakness (W1) is addressed with the added TabArena experiments. To me, they suggest a smaller gap between POT and TACO than the old experiments, but a larger gap to TabICL. However, much of this gap is probably explained by the use of a weaker open-source prior and small-data-only pre-training, and therefore does not suggest an inherent limitation of the method. I will raise my score to 5.
> >
> > Regarding the fit memory: Thanks for the explanation, I might have misread MB as GB in the fit memory column. But I still find it hard to make sense of some of the memory usages, for which it doesn't help that I don't know what train/test size and number of features was used in these tables. E.g. in Table 2, the peak predict memory should be from storing a few large tensors in a layer at the same time (e.g., residual, q, k, v), plus the sdpa memory. However, while the predictor in TACO can have lower memory usage, I would think that the compressor in TACO should have similarly large tensors as the predictor in POT, and therefore, POT and TACO should have similar memory usage? (Also, your answer about the 3x memory reduction references line 141 right, which is about runtime, not memory.) In Table 4 predict memory, why would TACO 16% be roughly halfway in-between TACO 8% and POT? This suggests that TACO 100% would need much more memory than POT, while to my understanding it seems that they should use roughly the same amount of memory. Also, in Table 4, why does POT use much less predict memory than fit memory? I thought the majority of memory should be consumed by the KV cache, which needs to be present at fit and predict time. (That is, unless the implementation somehow doesn't deallocate intermediate tensors after their use, e.g. because it doesn't use no_grad() or because variables are not deallocated.) For POT I would expect KV cache size = 12x{k_train,v_train} while intermediate tensors should be something like {residual, q, k, v, sdpa memory} which I guess should be smaller unless n_test >> n_train. (Sorry the answer is a bit unorganized because of a lack of time, but the main point is I would appreciate being able to understand better from the paper what memory usage to expect and why, and maybe to poke if there is an inefficiency in the implementation that would explain my confusion.)

---

> > > ### Author Response · Authors · 2026-04-02
> > >
> > > We thank the reviewer for the interesting questions. Below are our answers to the additional questions:
> > >
> > > ## On Table 2 predict memory, why TACO and POT should have similar peak memory:
> > >
> > > The reviewer's intuition is correct: during the first prediction, the compressor in TACO does process tensors of similar size as the predictor in POT. Based on the data, TACO's first-prediction peak memory ranges from 22.69 GB (1% compression) to 26.34 GB (16%), compared to POT's 22.45 GB.
> > >
> > > The slightly higher first-prediction memory for TACO is explained by the compressor's sequence length. Just as POT jointly processes train rows and test rows, TACO's compressor jointly processes train rows and the dummy rows. At 1% compression, that is 15,000 + 150 = 15,150 rows, compared to POT's 15,000 + 50 = 15,050. At 16%, the compressor processes 15,000 + 2,400 = 17,400 rows, explaining the larger 26.34 GB peak. The additional dummy rows increase the attention computation and intermediate tensor sizes proportionally.
> > >
> > > The difference in the reported mean/median arises because Table 2 averages over 100 sequential test batches. Without KV caching, POT must reprocess the full training context (n_train = 15,000, n_features = 500) for every test batch, so it pays 22.45 GB on every one of the 100 calls. TACO, by contrast, runs the compressor only on the first call, stores the resulting compressed context internally, and then subsequent calls (batches 2-100) run only the predictor on this much smaller compressed context. The subsequent prediction cost is 0.54 GB at 1% compression and 4.89 GB at 16%. We note that the synthetic dataset size used for this benchmark (n_train = 15,000, n_features = 500) is present in the main manuscript (line 260 right).
> > >
> > > To avoid confusions in the future, we will separate the first-prediction memory as a distinct column in Table 2, following the discussion with the reviewer as follows:
> > >
> > > https://anonymous.4open.science/r/taco-AAA1/rebuttal/times_and_memory_no_kv_cache.md
> > >
> > >
> > > ## On Table 4 predict memory, why TACO 16% appears halfway between TACO 8% and POT:
> > >
> > > Thank you, this comment helped us identify an implementation asymmetry in our cached-prediction benchmark. In the original Table 4 measurement, cached POT prediction offloaded each ensemble member from GPU after use, whereas cached TACO prediction kept all ensemble members resident on GPU simultaneously. This inflated TACO's reported peak predict memory, making the scaling with compression rate appear steeper than it should be. After matching the offloading behavior, this effect disappears, supporting the reviewer's intuition that a hypothetical TACO 100% should be in the same ballpark as POT rather than substantially larger. Below, we report both settings (with and without estimator offloading) to make this memory/time tradeoff explicit.
> > >
> > > No offloading:
> > > https://anonymous.4open.science/r/taco-AAA1/rebuttal/times_and_memory_kv_cache_no_offloading.md
> > >
> > > With offloading:
> > > https://anonymous.4open.science/r/taco-AAA1/rebuttal/times_and_memory_kv_cache_offloading.md
> > >
> > >
> > > ## On Table 4, why POT uses much less predict memory than fit memory:
> > >
> > > Thank you for pointing this out. For POT, this fit-versus-predict asymmetry is expected and is not due to missing no_grad() or because the tensors are not deallocated; our benchmark prediction calls are run under torch.inference_mode(). This behavior comes from the TabPFNv2 implementation that we use. The key point is that Table 4 reports peak memory during each call. During fit, POT constructs the KV cache by processing the full 15,000-row training sequence, so the peak includes both the cache being built and the large intermediate train-side attention tensors created during cache construction. During prediction, the KV cache is already stored, and POT only processes a small test batch of 50 rows while attending against the cached training keys and values. As a result, the intermediate query-side tensors are much smaller at prediction time, which is why peak prediction memory is substantially lower even though the KV cache itself remains in memory.
> > >
> > > We thank the reviewer for the additional questions. As a reminder, in case it was forgotten, we would like to kindly remind the reviewer for the score update.

---

### Official Review · Reviewer_eA7L · 2026-03-07

**Soundness:** 2
**Presentation:** 2
**Significance:** 1
**Originality:** 2
**Overall Recommendation:** 3
**Confidence:** 5

**Summary:**

The paper introduces TACO, a compression method designed for tabular in-context learning models. Tabular foundation models typically scale quadratically or linearly with the number of training instances during inference. To address this, TACO utilizes a compressor network to map the training dataset into a smaller latent representation. A separate predictor network then uses this compressed representation to make predictions on test data. The authors evaluate the method on the TabArena benchmark to measure changes in inference speed, memory consumption, and predictive performance across various compression rates.

**Compliance With Llm Reviewing Policy:**

Affirmed.

**Final Justification:**

I thank the authors for their response and am raising my score to a 3. I appreciate the new empirical results. However, I am keeping my score on the rejection side because a method claiming to advance Tabular Foundation Models remains fundamentally incomplete without evaluating regression tasks, which make up a massive portion of real-world tabular problems. While I understand that compute constraints prevented training a regression backbone during the rebuttal window, this missing capability and the initial omission of a dedicated limitations section mean the proposed framework requires further expansion before it is ready for publication.

**Key Questions For Authors:**

1. Can you provide empirical results for regression datasets to verify that the latent compression retains continuous target relationships?

2. How does TACO perform against proper dataset distillation or coreset selection techniques rather than just random sampling?

3. Have you tested alternative initialization strategies for the dummy rows instead of random selection from the training set?

4. How does the compressor module handle extreme class imbalance within the chunking and stitching framework?

**Limitations:**

No. The authors provide an Impact Statement but fail to explicitly discuss the technical limitations of their work, such as the reliance on synthetic pretraining data or the lack of regression evaluation. They need to add a dedicated limitations section.

**Strengths And Weaknesses:**

Soundness: The technical approach is logically structured, separating the compression and prediction phases. The evaluation uses appropriate metrics like ROC AUC and incorporates critical difference diagrams for statistical validation. However, evaluating exclusively on the TabArena classification datasets limits the generalizability of the empirical claims, as regression tasks are omitted.

Presentation: The writing is clear, and the architecture diagrams adequately explain the flow of tensors between the compressor and predictor modules. The motivation regarding the hardware constraints of large-scale tabular attention is well established. Some algorithm details regarding the exact formulation of the dummy rows could be expanded for better reproducibility.

Significance: Reducing the memory footprint and latency of tabular foundation models is an important practical problem for deployment. The reported memory savings and speedups present a clear utility for handling larger context sizes.
Originality: The idea of compressing the context into a fixed-size latent buffer is an interesting application of end-to-end representation learning for tabular foundation models. While context compression exists in natural language processing, adapting it specifically to tabular in-context learning with a dual network setup offers a distinct perspective.

## Strengths:

- The dual module architecture effectively decouples the context size from the prediction complexity, enabling substantial reductions in inference latency and memory.

- The proposed chunking and stitching strategy provides a highly practical mechanism to scale transformer-based tabular models to datasets with over one million rows.

## Weaknesses/Suggestions:

- The evaluation is restricted to classification tasks within the TabArena benchmark. Extending the experiments to regression tasks would provide a more comprehensive assessment of the predictive capabilities of the compressed representations.

- The baseline comparisons are limited to simple KNN and random sampling. Comparing against other efficient architectures or distillation methods would strengthen the empirical claims.

- The paper relies exclusively on synthetically generated datasets using structural causal models for pretraining. It is unclear how the compressor would perform if pretrained on real-world noisy tabular data.

- The dummy table rows used to initialize the compressor are selected randomly from the training set. The authors should investigate whether a more deterministic or clustering-based initialization improves the stability of the learned latent space.

---

> ### Author Rebuttal · Authors · 2026-03-30
>
> We thank the reviewer for the detailed feedback. We address the main concerns from the reviewer below:
>
> ## Regarding the regression evaluation:
> We refer the reviewer to our response to Reviewer 9YHr for a detailed discussion on regression evaluation.
>
> ## Regarding additional distillation baselines
> We thank the reviewer for the valuable suggestion. We would like to note that random sampling and kNN-like sampling are the de facto adopted strategies in the community for running TabPFN on large datasets Ye et al. (2025, arXiv:2502.17361).
>
> In terms of distillation/efficient baselines, we have identified the following:
> - TabFlex
> - MotherNet
> - HyperFast
>
> For the comparison with TabFlex, we would kindly point the reviewer to the following figure, where our method outperforms TabFlex:
>
> https://anonymous.4open.science/r/taco-AAA1/rebuttal/full_tabarena_elo_results.png
>
> For MotherNet it is unfortunately not possible to run it as the method checkpoint is not available anymore from the authors. We have additionally contacted the authors but have received no reply.
>
> HyperFast is not present in TabArena, so we ran the method ourselves on the same benchmark, and we provide the results at the next **question related to synthetic data pretraining.**
>
> Based on the results, we confirm that TACO still manages to outperform related distillation and efficient approaches.
>
> ## Regarding synthetic data pretraining:
> We would like to point out to the reviewer that training on synthetic data is not a weakness, since synthetic data can be generated in abundance. The performance of our method has been evaluated in real datasets and it generalizes very well.
>
> Additionally, pretraining on synthetic data generated from SCMs is the established methodology for training state-of-the-art tabular foundation models such as TabPFN, TabPFNv2, TabPFN2.5, TabICL, TabICLv2, Mitra.
>
> That said, based on RealTabPFN, we have conducted additional experiments with finetuning TACO and POT on real data.
>
> | Method | Score |
> |---|---:|
> | POT (finetuned) | 0.862 ± 0.101 |
> | POT | 0.859 ± 0.102 |
> | TACO (finetuned, 16%) | 0.858 ± 0.100 |
> | TACO (finetuned, 8%) | 0.858 ± 0.100 |
> | TACO (finetuned, 4%) | 0.857 ± 0.099 |
> | TACO (finetuned, 2%) | 0.857 ± 0.098 |
> | TACO (finetuned, 1%) | 0.855 ± 0.098 |
> | TACO (16%) | 0.855 ± 0.100 |
> | TACO (8%) | 0.855 ± 0.099 |
> | TACO (4%) | 0.855 ± 0.099 |
> | TACO (2%) | 0.854 ± 0.099 |
> | TACO (1%) | 0.854 ± 0.098 |
> | HyperFast | 0.8224 ± 0.106 |
>
> As can be observed, training on real data improves generalization.
>
> ## Dummy Row Initialization
> To directly address the reviewer’s suggestion, we trained TACO from scratch for only 5K steps with our initial strategy, and with using k-mean centroids as dummy rows.
>
> We observe that k-means initialization introduces approximately 5 seconds of additional overhead in the first prediction time.
>
> In terms of predictive quality (measured on TabArena under the same setting as Table 1 in the paper), the baseline TACO achieves a score of 0.786, while the k-means initialized variant achieves 0.781, indicating a slight degradation with clustering-based initialization in this setting. We also observe that the training losses of the two models are very close, suggesting similar optimization behavior. However, the k-means variant incurs a higher training cost, with total training time increasing to approximately 150% of the baseline.
>
> Given these results, clustering-based initialization does not provide a clear benefit in our setting, while introducing additional computational overhead at both training and inference time. While alternative clustering strategies or larger-scale experiments may yield different outcomes, we do not find sufficient evidence to justify its use over simpler initialization schemes.
>
> ## Regarding class imbalance in chunking
> The compressor learns to distill the informational content of each chunk into a compressed representation, it does not subsample rows. This means that even if a rare class appears in only a single chunk, that signal is encoded into the compressed context for that chunk and preserved when chunks are stitched together. This is a key advantage of learned compression over methods like random sampling or kNN, where rare class instances can simply be dropped. Additionally, the chunking procedure is compatible with stratified sampling strategies if further control over class representation within chunks is desired.
>
> Our MetroPT-3 experiment (Table 3) demonstrates this empirically under extreme class imbalance (0.1% anomaly rate), where TACO outperforms all compression baselines on AUPRC and the newly added F1 score (Reply to reviewer WdoC).
>
> ## Regarding the limitations section
> We refer the reviewer to our response to Reviewer 9YHr for a discussion on the limitations section.
>
> We believe we have addressed all the concerns raised by the reviewer. We would kindly ask the reviewer to increase the score and recommend acceptance based on the provided clarifications.

---

> > ### Author Rebuttal · Reviewer_eA7L · 2026-04-03
> >
> > I thank the authors for their response and am raising my score to a 3. I appreciate the new empirical results. However, I am keeping my score on the rejection side because a method claiming to advance Tabular Foundation Models remains fundamentally incomplete without evaluating regression tasks, which make up a massive portion of real-world tabular problems. While I understand that compute constraints prevented training a regression backbone during the rebuttal window, this missing capability and the initial omission of a dedicated limitations section mean the proposed framework requires further expansion before it is ready for publication.

---

> > > ### Author Response · Authors · 2026-04-03
> > >
> > > We would like to thank the reviewer for the interesting discussion and for increasing the score.
> > >
> > > Regarding regression results, we can unfortunately not obtain it within the rebuttal period, however, we would again like to point out that state-of-the-art open-source accepted work does not provide a regression backbone [1].
> > >
> > > Regarding the limitations section: given the page constraints, we prioritized including additional experimental insights and ablations. We will add a dedicated limitations section in the revision covering the current restriction to classification.
> > >
> > > **Edit: 07.04.2026**
> > >
> > > That said, we fully agree that explicitly discussing limitations is important. To address this, we provide a draft below and will include a dedicated limitations section in the camera-ready version:
> > >
> > > **Even though TACO advances the field of Tabular Foundation Models by achieving scalability to millions of training data points, it can still be improved along several dimensions. In particular, pretraining could benefit from more diverse priors that better capture real-world datasets, including those with a higher degree of missing values. Furthermore, incorporating datasets that reflect distribution shifts would be important for real-world deployment scenarios, recent work [1] has shown that temporal distribution shift handling can be integrated into the TFM regime, and a similar approach could be applied to TACO. While we benchmark TACO on established tabular benchmarks following prior work [2,3,4], we acknowledge that our evaluation is currently limited to classification tasks. Extending to regression and time-series settings remains an important direction for future work.**
> > >
> > >
> > > [1] Helli, K., Schnurr, D., Hollmann, N., Müller, S., & Hutter, F. (2024). Drift-Resilient TabPFN: In-Context Learning Temporal Distribution Shifts on Tabular Data. NeurIPS.
> > >
> > > [2] Qu, J., Holzmüller, D., Varoquaux, G., & Le Morvan, M. (2025). TabICL: A Tabular Foundation Model for In-Context Learning on Large Data. ICML.
> > >
> > > [3] Hollmann, N., Müller, S., Purucker, L., Krishnakumar, A., Körfer, M., Hoo, S. B., et al. (2025). Accurate predictions on small data with a tabular foundation model. Nature, 637(8045), 319–326.
> > >
> > > [4] Zhang, X., Maddix, D. C., Yin, J., Erickson, N., Ansari, A. F., Han, B., et al. (2025). Mitra: Mixed Synthetic Priors for Enhancing Tabular Foundation Models. NeurIPS.

---

### Official Review · Reviewer_WdoC · 2026-03-09

**Soundness:** 3
**Presentation:** 3
**Significance:** 3
**Originality:** 4
**Overall Recommendation:** 5
**Confidence:** 4

**Summary:**

This paper proposes scaling tabular foundation models by decomposing it into a compression stage and a prediction stage, both being TabPFNs, so that it can "compress once, predict many times" with a much larger throughput and much faster speed.

**Compliance With Llm Reviewing Policy:**

Affirmed.

**Final Justification:**

After reading the reviewer's comments and the reply to acknowledgement, I have raised my score to 5.

**Key Questions For Authors:**

See weaknesses.

**Limitations:**

See weaknesses.

**Strengths And Weaknesses:**

Strengths:
* Scaling tabular foundation models is a real problem, and this paper provided a simple yet elegant solution.
  * Similar ideas include (1) TuneTables and ICD, both distilling the training set into smaller sizes (prompt tokens or dummy rows), but require one distill per dataset, (2) MotherNet that takes in the training set once, outputs an MLP that can be used multiple times for prediction, but not scaling as well as this paper did.  In particular, Hypothesis 6 is very impressive - the composability of compressed chunks made it among the first few works that scaled TabPFN to millions (together with TabICLv2 [1] and TabFlex [2], although [1] is concurrent work and [2] uses linear attention).
* Presentation is clean.  Very easy to understand.
* Ablation studies are thorough.

Weaknesses:

* The only major concern I have on the soundness is the lack of comparison against established baselines.  Most of the experiments regarding the prediction quality (Table 1, 3, 6, 7) are only comparing against POT (prediction-only predictor) as well as different choices of core sets (learned vs random vs kNN), with the exception of Table 1 that additionally compares against TabPFNv2 and TabICL.  How do TabPFNv2/v2.5, TabICL, and supervised methods (such as XGBoost/AutoGluon) perform on these datasets?  Namely, even if TabPFN cannot run on millions of rows, XGBoost/AutoGluon could.  What are their prediction quality as well as their fit/time-to-first-predict/predict latency?  They remain to be seen, and they are essential to evaluate the practicality of this approach on large datasets.
* A small weakness on MetroPT-3: MetroPT-3 is an anomaly detection dataset, where only 0.1% are anomalies.  Accuracy and even AUROC would be deceivingly high.  AUPRC, F1 score, etc., could be more meaningful.  Besides, [3] showed that kNN could achieve 75.2 F1 score, and their method achieving 99.0% F1 score (see Table 5 in their paper).
   * If TACO is genuinely stronger than their model, then a catchier paper title could be something like "Scaling tabular foundation models to a million rows (or tens of millions of cells) in xx seconds".

[1] TabICLv2: A better, faster, scalable, and open tabular foundation model, Qu et al., 2026
[2] TabFlex: Scaling Tabular Learning to Millions with Linear Attention, Zeng et al., 2025
[3] IntelliMetro-Hybrid: A Machine Learning and Deep Learning Fusion Model for Economic Optimization in Smart Metro Systems, Peng et al., 2025

---

> ### Author Rebuttal · Authors · 2026-03-29
>
> We thank the reviewer for the thoughtful evaluation and for recognizing the novelty and elegance of our approach. Below we address the main concerns from the reviewer:
>
> ## Regarding baseline performances:
> We thank the reviewer for raising an interesting point. Given that we run on TabArena, we pulled the leaderboard state with both POT and TACO present at the moment of writing this response.
>
> We present the results in the following figure:
>
> https://anonymous.4open.science/r/taco-AAA1/rebuttal/full_tabarena_elo_results.png
>
> The leaderboard state is consistent with the results presented in our work. We have open-sourced the implementation and model weights, and the results can be verified by pulling the leaderboard state locally.
>
> Additionally, we wanted to make the following remarks:
> - Our method uses the TabPFNv2 architecture.
> - We do not have access to the TabPFNv2, Mitra, or TabICLv2 prior.
> - For the current results, we have additionally trained our method on more synthetic, and lastly, real datasets. However, we still have only run for up to 91k steps, while for e.g. TabPFNv2 has run for 130k steps. One single training run of our method for 91k steps takes approximately 18 days on 8×H100 GPUs.
>
> ## Regarding baseline fit times:
> As the reviewer suggests we provide the fit and inference times of XGBoost and AutoGluon in Table 2. Tree-based methods are still faster in our experiments compared to Tabular Foundation Models, however, our method has a better training and inference time compared to AutoGluon (training time is an argument, we select a common runtime of 30 min). These results are consistent with the time statistics from the official TabArena leaderboard.
>
> *Note:* Parentheses indicate × speed-up and -% memory reduction versus POT.
> | Method | Fit Time | First Predict Time | Subsequent Predict Time (Mean ± Std) | Fit Mem | Predict Mem Mean | Predict Mem Median |
> |---|---:|---:|---:|---:|---:|---:|
> | POT | 8.41s | 28.55s | 28.67s ± 0.05s | 36MB | 22.45GB | 22.45GB |
> | TACO 1% | 8.22s | 29.68s | 306ms ± 18ms (×93.6 speed-up) | 92MB | 776MB (-96.6%) | 549MB (-97.6%) |
> | TACO 2% | 8.30s | 29.80s | 382ms ± 18ms (×75.2 speed-up) | 92MB | 1.05GB (-95.3%) | 845MB (-96.3%) |
> | TACO 4% | 9.86s | 30.71s | 544ms ± 17ms (×52.7 speed-up) | 92MB | 1.63GB (-92.8%) | 1.41GB (-93.7%) |
> | TACO 8% | 10.02s | 32.47s | 943ms ± 17ms (×30.4 speed-up) | 92MB | 2.78GB (-87.6%) | 2.56GB (-88.6%) |
> | TACO 16% | 9.20s | 35.36s | 1.91s ± 0.01s (×15 speed-up) | 92MB | 5.10GB (-77.3%) | 4.89GB (-78.2%) |
> | XGBoost | 3.14s | 154ms | 6.3ms ± 0.2ms | 8MB | 8MB | 8MB |
> | AutoGluon Extreme (30m) | 30.2min | 1.32s | 947ms ± 10ms | 6.41GB | 8MB | 8MB |
>
> ## Regarding MetroPT-3:
> We wanted to kindly point out that the related work that the reviewer cites, runs on the same task, but on a different target variable compared to ours. Secondly, the work that the reviewer cites is a system of 2 phases, where first tree-based methods are applied, which are then combined with deep neural networks. As such, we believe comparing to systems that apply multiple methods is out of scope for our work, especially since our focus is not on obtaining state-of-the-art accuracy.
>
> The aim of the experiment is to verify that via our proposed approach, one could close the performance gap for medium to large scale datasets between state-of-the-art methods (such as XGBoost and AutoGluon that the reviewer cites) and tabular foundation models that have a limited context.
>
> To further strengthen the results, we added TabPFNv2 with random sampling and TabPFNv2 with KNN-based sampling to the experiment. We additionally added XGBoost as a reference, and AutoGluon. Given that AUPRC was already present in the table, following the reviewer’s suggestion we added precision, recall, and F1.
>
> Results on MetroPT-3 (mean ± std over 5 seeds)
>
> | Method | Precision | Recall | F1 | LogLoss | ROC-AUC | AUPRC |
> |---|---:|---:|---:|---:|---:|---:|
> | TACO (chunk) | **0.989 ± 0.009** | 0.792 ± 0.071 | 0.878 ± 0.043 | 0.005 ± 0.001 | 0.984 ± 0.011 | 0.896 ± 0.008 |
> | POT (Random) | 0.636 ± 0.353 | 0.689 ± 0.075 | 0.630 ± 0.233 | 0.028 ± 0.012 | 0.924 ± 0.042 | 0.729 ± 0.112 |
> | POT (kNN) | 0.495 ± 0.445 | 0.317 ± 0.305 | 0.172 ± 0.141 | 0.840 ± 1.190 | 0.903 ± 0.108 | 0.372 ± 0.344 |
> | TabPFNv2 (Random) | 0.616 ± 0.415 | 0.733 ± 0.110 | 0.606 ± 0.326 | 0.030 ± 0.012 | 0.948 ± 0.029 | 0.744 ± 0.118 |
> | TabPFNv2 (kNN) | 0.187 ± 0.252 | 0.341 ± 0.309 | 0.128 ± 0.105 | 1.174 ± 1.654 | 0.905 ± 0.107 | 0.236 ± 0.204 |
> | AutoGluon (extreme, 30m) | 0.982 ± 0.001 | **0.889 ± 0.003** | **0.933 ± 0.001** | 0.003 ± 0.000 | 0.999 ± 0.000 | **0.966 ± 0.002** |
> | XGBoost | 0.972 ± 0.002 | 0.847 ± 0.036 | 0.905 ± 0.021 | **0.003 ± 0.001** | **0.999 ± 0.000** | 0.959 ± 0.007 |
>
> We believe to have addressed all the concerns raised by the reviewer. If the reviewer has additional concerns we are happy to answer them.

---

> > ### Author Rebuttal · Reviewer_WdoC · 2026-04-03
> >
> > Most of the questions are addressed, although TACO/POT is in general worse than TabICL/Mitra/TabPFNv2, as well as XGBoost/AutoGluon on MetroPT-3.  Therefore I choose to maintain my weak accept score, partially for encouraging this attempt to scale TabPFN-like models, but also acknowledging the gap towards SOTA.

---

> > > ### Author Response · Authors · 2026-04-03
> > >
> > > We would like to thank the reviewer for the valuable discussion. We would like to point out that achieving SOTA performance was not a target of our work.
> > >
> > > If we could make a parallelisation, there are multiple efficient techniques being proposed for example in the natural language domain for foundation models and the results provided are compared to models trained for a limited budget. The results are not compared to the latest version of ChatGPT or Claude.
> > >
> > > Additionally, AutoGluon by itself uses TabPFN like tabular foundation models in it’s ensemble, as such we believe a single model vs an ensembling/stacking solution of various models (including the same architecture) is bound to outperform it.
> > >
> > > If the aforementioned argument satisfies the reviewer we would appreciate additional support with the score. Otherwise, we respect the reviewers decision and we thank the reviewer for the discussion.

---

### Decision · Program_Chairs · 2026-04-30

**Decision:**

Accept (spotlight)

**Comment:**

This paper tackles an important problem in an intuitive way. It is well-written with thorough ablations and is open-sourced. It is a straightforward but original method. Concerns about evaluations against classical benchmarks have been addressed in the rebuttal period. There remains the limitation that this technique is not yet proven to work for regression, but I consider this to be future work as opposed to a reason for rejection.

I expect that the authors will make changes promised in the rebuttal in time for the camera-ready deadline.